# Determination of vadose and saturated-zone nitrate lag times using long-term groundwater monitoring data and statistical machine learning

Martin J. Wells[1,3], Troy E. Gilmore[2,3], Natalie Nelson[4,5], Aaron Mittelstet[3], J.K. Böhlke[6],

[1]currently at Natural Resources Conservation Service, Redmond, OR, 97756, USA

[2]Conservation and Survey Division - School of Natural Resources, University of Nebraska, Lincoln, NE, 68583, USA

[3]Biological Systems Engineering, University of Nebraska, Lincoln, NE, 68583, USA

[4]Biological and Agricultural Engineering, North Carolina State University, Raleigh, NC, 27695, USA

[5]Center for Geospatial Analytics, North Carolina State University, Raleigh, NC, 27695, USA

[6]U.S. Geological Survey, Reston, VA, 20192, USA

*Correspondence to*: Troy E. Gilmore (gilmore@unl.edu)

**Abstract.** In this study, we explored the use of statistical machine learning and long-term groundwater nitrate monitoring data to estimate vadose-zone and saturated-zone lag times in an irrigated alluvial agricultural setting. Unlike most previous statistical machine learning studies that sought to predict groundwater nitrate concentrations within aquifers, the focus of this study was to leverage available groundwater nitrate concentrations and other environmental variables to determine mean regional vertical velocities (transport rates) of water and solutes in the vadose zone and saturated zone (3.50 m/year and 3.75 m/year, respectively). The statistical machine learning results are consistent with two primary recharge processes in this Western Nebraska aquifer: (1) diffuse recharge from irrigation and precipitation across the landscape, and (2) focused recharge from leaking irrigation conveyance canals. The vadose-zone mean velocity yielded a mean recharge rate (0.46 m/year) consistent with previous estimates from groundwater age-dating in shallow wells (0.38 m/year). The saturated zone mean velocity yielded a recharge rate (1.31 m/year) that was more consistent with focused recharge from leaky irrigation canals, as indicated by previous results of groundwater age-dating in intermediate-depth wells (1.22 m/year). Collectively, the statistical machine-learning model results are consistent with previous observations of relatively high-water fluxes and short transit times for water and nitrate in the primarily oxic aquifer. Partial dependence plots from the model indicate a sharp threshold where high groundwater nitrate concentrations are mostly associated with total travel times of seven years or less, possibly reflecting some combination of recent management practices and a tendency for nitrate concentrations to be higher in diffuse infiltration recharge than in canal leakage water. Limitations to the machine learning approach include non-uniqueness of different transport rate combinations when comparing model performance and highlight the need to corroborate statistical model results with a robust conceptual model and complementary information such as groundwater age.

## 1 Introduction

Nitrate is a common contaminant of groundwater and surface water that can affect drinking water quality and ecosystem health. Predicting responses of aquatic resources to changes in nitrate loading can be complicated by uncertainties related to rates and pathways of nitrate transport from sources to receptors. Lag times for movement of non-point source nitrate contamination through the subsurface are widely recognized (Böhlke, 2002; Meals et al., 2010; Puckett et al., 2011; Van Meter and Basu, 2017) but difficult to measure. Vadose (unsaturated zone) and groundwater (saturated zone) lag times are of critical importance for monitoring, regulating, and managing the transport of contaminants in groundwater. However, transport time-scales are often generalized due to coarse spatial and temporal resolution in data available for groundwater systems impacted by agricultural activities (Gilmore et al., 2016; Green et al., 2018; Puckett et al., 2011), resulting in a simplified groundwater management approach. Regulators and stakeholders in agricultural landscapes are increasingly in need of more precise and local lag time information to better evaluate and apply regulations and best management practices for the reduction of groundwater nitrate concentrations (e.g., Eberts et al., 2013).

Field-based studies of lag times (time to move through both vadose zone and aquifer) commonly use vadose-zone sampling and/or expensive groundwater age-dating techniques to estimate nitrate transport rates moving into and through aquifers (Böhlke et al., 2002, 2007; Böhlke and Denver, 1995; Browne and Guldan, 2005; Kennedy et al., 2009; McMahon et al., 2006; Morgenstern et al., 2015; Turkeltaub et al., 2016; Wells et al., 2018). Detailed process-based modelling studies focused on lag times require complex numerical models combined with spatially intensive and/or costly hydrogeological observations (Ilampooranan et al., 2019; Rossman et al., 2014; Russoniello et al., 2016). Thus, efficient but locally-applicable modelling approaches are needed (Green et al., 2018; Liao et al., 2012; Van Meter and Basu, 2015). In this study, an alternative data-driven approach (Random Forest Regression) leverages existing long-term groundwater nitrate concentration (referred to as [$NO_3^-$] hereafter) data and easily accessible environmental data to estimate vadose and saturated-zone vertical velocities (transport rates) for the determination of subsurface lag times.

Statistical machine learning methods, including Random Forest, have been used successfully for modelling [$NO_3^-$] distribution in aquifers (Anning et al., 2012; Juntakut et al., 2019; Knoll et al., 2020; Nolan et al., 2014; Ouedraogo et al., 2017; Rodriguez-Galiano et al., 2014; Rahmati et al., 2019; Vanclooster et al., 2020; Wheeler et al., 2015), but there has not been robust analysis of model capabilities for estimating vadose and/or saturated-zone lag times. Proxies for lag time, such as well screen depth, have been used as predictors in Random Forest models (Nolan et al., 2014; Wheeler et al., 2015). Decadal lag times have been suggested from using time-averaged nitrogen inputs as predictors (e.g., 1978-1990 inputs vs 1992-2006 inputs) and by comparing their relative importance in the model (Wheeler et al., 2015). Application of similar machine learning methods suggested groundwater age could be used as a predictor to improve model performance (Ransom et al., 2017). Hybrid models, using both mechanistic models and machine learning, have also sought to integrate vertical transport model parameters and outputs to evaluate nitrate-related predictors, including vadose-zone travel times (Nolan et al., 2018).

The objective of this study is to test a data-driven approach for estimating vadose and saturated-zone transport rates and lag times for an intensively monitored alluvial aquifer in western Nebraska (Böhlke et al., 2007; Verstraeten et al., 2001a, 2001b; Wells et al., 2018). Results are compared to the hydrogeologic, mechanistic understanding from previous groundwater studies to determine strengths and weaknesses of the approach as (1) a stand-alone technique, or (2) as an exploratory analysis to guide or complement more complex physical-based models or intensive hydrogeologic field investigations.

## 2 Methods

### 2.1 Site Description

The Dutch Flats study area is in the western Nebraska counties of Scotts Bluff and Sioux (Fig. 1). The North Platte River delivers large quantities of water for crop irrigation in this region and runs along the southern portion of this study area. Irrigation water is diverted from the North Platte River into three major canals (Mitchell-Gering, Tri-State, and Interstate Canals) that feed a network of minor canals. Several previous Dutch Flats area studies have investigated groundwater characteristics and provided thorough site descriptions of the semi-arid region (Babcock et al., 1951; Böhlke et al., 2007; Verstraeten et al., 2001a, 2001b; Wells et al., 2018). The Dutch Flats area overlies an alluvial aquifer characterized by unconsolidated deposits of predominantly sand and gravel, with the aquifer base largely consisting of consolidated deposits of the Brule, Chadron, or Lance Formation (Verstraeten et al., 1995) (Fig. 2). Irrigation water not derived from the North Platte River is typically pumped from the alluvial aquifer, or water-bearing units of the Brule Formation.

The total area of the Dutch Flats study area is roughly 540 km$^2$, of which approximately 290 km$^2$ (53.5%) is agricultural land (cultivated crops and pasture). Most agricultural land is concentrated south of the Interstate Canal (Homer et al., 2015). Due to the combination of intense agriculture and low annual precipitation, producers in Dutch Flats rely on a network of irrigation canals to supply water to the region. From 1908 to 2016, mean precipitation of 390 mm was measured at the nearby Western Regional Airport in Scottsbluff, NE (NOAA, 2017).

While some groundwater is withdrawn for irrigation, and some irrigated acres in the study area are classified as commingled (groundwater and surface water source), Scotts Bluff County irrigation is mostly from surface water sources. Estimates determined every five years suggest surface water provided between 76.8% to 98.6% of the total water withdrawals from 1985 to 2015, or about 92% on average (Dieter et al., 2018). Canals transport water from the North Platte River to fields throughout the study area, most of which are downgradient (south) of the Interstate Canal. Mitchell-Gering, Tri-State, and Interstate Canals are the major canals in Dutch Flats, with the latter holding the largest water right of 44.5 m$^3$/s (NEDNR, 2009). Leakage from these canals provides a source of artificial groundwater recharge. Previous studies estimate the leakage potential of canals in the region results in as much as 40% to 50% of canal water being lost during conveyance (Ball et al., 2006; Harvey and Sibray, 2001; Hobza and Andersen, 2010; Luckey and Cannia, 2006). Leakage estimates from a downstream section of the Interstate Canal (extending to the east of the study area; Hobza and Andersen (2010)) suggest fluxes ranging from 0.08 to 0.7 m day$^{-1}$ through the canal bed. Assuming leakage of 0.39 m day$^{-1}$ over the Interstate Canal bed area (16.8 m width x 55.5 km length) within Dutch Flats yields 4.1 x 10$^5$ m$^3$ day$^{-1}$ of leakage. Applied over an on-average 151-day operation period (USBR, 2018), leakage from Interstate Canal alone could approach 6.1 x 10$^7$ m$^3$ annually, or about 29% of the annual volume of precipitation in the Dutch Flats area.

A 1990s study investigated both spatial and temporal influences from canals in the Dutch Flats area (Verstraeten et al., 2001a, 2001b), with results later synthesized by Böhlke et al. (2007). Canals were found to dilute groundwater [NO$_3^-$] locally with low-[NO$_3^-$] (e.g., [NO$_3^-$] < 0.06 mg N L$^{-1}$ in 1997) surface water during irrigation season. $^3$H/$^3$He age-dating was used to determine apparent groundwater ages and recharge rates. It was noted that wells near canals displayed evidence of high recharge rates influenced by local canal leakage. Data from wells far from the canals indicated that shallow groundwater was more likely influenced by local irrigation practices (i.e., furrows in fields), while deeper groundwater was impacted by both localized irrigation and canal leakage (Böhlke et al., 2007). Shallow groundwater in the Dutch Flats area has hydrogen and oxygen stable isotopic compositions consistent with surface water sources (i.e., North Platte River and associated canals), indicating that most groundwater intercepted by the monitoring well network has been affected by surface-water irrigation recharge (Böhlke et al., 2007; Cherry et al., 2020).

The Dutch Flats area is within the North Platte Natural Resources District (NPNRD), one of 23 groundwater management districts in Nebraska tasked with, among other functions, improving water quality and quantity. The NPNRD has a large monitoring well network consisting of 797 wells, 327 of which are nested. Nested well clusters are drilled and constructed such that screen intervals represent (1) "shallow" groundwater intersecting the water table (length of screened interval = 6.1 m), (2) "intermediate" groundwater from mid-aquifer depths (length of screened interval = 1.5 m), and "deep" groundwater near the base of the unconfined aquifer (length of screened interval = 1.5 m). Depending on well location within the Dutch Flats area, depths of the water table and base of aquifer are highly variable, such that shallow, intermediate, and deep wells can have overlapping ranges of depths below land surface (Fig. 2).

Influenced by both regulatory and economic incentives, the Dutch Flats area has undergone a notable shift in irrigation practices in the last two decades. From 1999 to 2017, center pivot irrigated area has increased by approximately 270%, from roughly 3,830 hectares to 14,253 hectares, or from 13% to 49% of the total agricultural land area, respectively. Most of this shift in technology has occurred on fields previously under furrow irrigation. Conventional furrow irrigation has an estimated potential application efficiency ("measure of the fraction of the total volume of water delivered to the farm or field to that which is stored in the root zone to meet the crop evapotranspiration needs," per Irmak et al. (2011)) of 45% to 65%, compared to center pivot sprinklers at 75% to 85% (Irmak et al., 2011). Based on improved irrigation efficiency (between 10-40%), average precipitation throughout growing season (29.5 cm for 15 April to 13 October (Yonts, 2002)), and average water requirements for corn (69.2 cm (Yonts, 2002)), converting furrow irrigated fields to center pivot over the aforementioned 14,253 hectares could represent a difference of $1 \times 10^7$ m$^3$ to $6 \times 10^7$ m$^3$ in water applied. Those (roughly approximated) differences in water volumes are equivalent to 6% to 28% of average annual precipitation applied over the Dutch Flats area, suggesting the change in irrigation practice does have potential to alter the water balance in the area.

The hypothesis of lower recharge due to changes in irrigation technology was investigated by Wells et al. (2018) by comparing samples collected in 1998 and 2016. Sample sites were selected based on a well's proximity to fields that observed a conversion in irrigation practices (i.e., furrow irrigation to center pivot) between the two collection periods. While mean recharge rate was not significantly different, a lower recharge rate was indicated by data from 88% of the wells. Long-term Dutch Flats [NO$_3^-$] trends were also assessed in the study, suggesting decreasing trends (though statistically insignificant) from 1998 to 2016 throughout the Dutch Flats area, and nitrogen isotopes of nitrate indicated little change in biogeochemical processes. For additional background, Wells et al., (2018) provides a more in-depth analysis of recent [NO$_3^-$] trends in this region (see also, Fig. S1A in the online Supplemental Material, which shows the nitrate data used in the present study).

As in other agricultural areas, nitrate in Dutch Flats groundwater is dependent on nitrogen loading at the land surface, rate of leaching below crop root zones, rate of nitrate transport through the vadose and saturated zones, dilution from focused recharge in the vicinity of canals, rate of discharge from the aquifer (whether from pumping or discharge to surface water bodies), and rates of nitrate reduction (primarily denitrification) in the aquifer. Based on nitrogen and oxygen isotopes in nitrate and redox conditions observed in previous studies, denitrification likely has a relatively minor or localized influence on groundwater nitrate in the Dutch Flats area (Wells et al., 2018). Evidence of denitrification (from dissolved gases and isotopes (Böhlke et al., 2007, Wells et al. 2018)) was mostly limited to some of the deepest wells near the bottom of the aquifer. Leakage of low-nitrate water in the major canals causes nitrate dilution in the groundwater (i.e., relatively little nitrate addition, at least from the upgradient canals). Additional isotope data might be useful for documenting temporal shifts in recharge sources, or irrigation return flows to the river; however, it is difficult to know exactly the location or size of the contributing area for each well, especially the deeper ones.

Other long-term changes to the landscape were evaluated by Wells et al. (2018) and included statistically significant reductions in mean fertilizer application rates (1987–1999 vs. 2000–2012) and volume of water diverted into the Interstate Canal

(1983–1999 vs. 2000–2016), while a significant increase in area of planted corn occurred (1983–1999 vs. 2000–2016).
Precipitation was also evaluated, and though the mean has decreased over a similar time period, the trend was not statistically
significant.
**2.2 Statistical Machine Learning Modelling Framework**
Statistical machine learning uses algorithms to assess and identify complex relationships between variables. Learned
relations can be used to uncover nonlinear trends in data that might otherwise be overshadowed when using simple regression
techniques (Hastie et al., 2009). In this study we used Random Forest Regression to evaluate site-specific explanatory variables
(e.g., precipitation, vadose-zone thickness, depth to bottom of screen, etc.) that may impact the response variable, groundwater
[$NO_3^-$]. Additionally, as described in detail in Section 2.4, we estimated a range of total travel times (from land surface to the point
of sampling) at each of the wells by varying vadose and saturated-zone transport rates. The relative importance of total travel time
as a predictor variable was ultimately used to identify an optimal travel time and model.
**2.3 Variables and Project Setup**
Data from 15 predictors were collected and analysed (Table 1). Spatial variables were manipulated using ArcGIS 10.4.
The [$NO_3^-$] dataset for the entire NPNRD had 10,676 observations from 1979 to 2014, and was downloaded from the Quality-
Assessed Agrichemical Contaminant Database for Nebraska Groundwater (University of Nebraska-Lincoln, 2016). We used data
encompassed by the Dutch Flats model area (2,829 [$NO_3^-$] observations from 214 wells). In order to have an accurate vadose-zone
thickness, only wells with a corresponding depth to groundwater record, of which the most recent record was used, were selected
(2,651 observations from 172 wells). Over this period, several wells were sampled much more frequently than others (e.g., monthly
sampling, over a short period of record), especially during a U.S. Geological Survey (USGS) National Water-Quality Assessment
(NAWQA) study from 1995 to 1999. To prevent those wells from dominating the training and testing of the model, annual median
[$NO_3^-$] was calculated for each well and used in the dataset. The dataset was further manipulated such that each median [$NO_3^-$]
observation had 15 complementary predictors (Table 1). The selected predictor variables capture drivers of long-term [$NO_3^-$] and
[$NO_3^-$] lags. After incorporating all data, including limited records of dissolved oxygen (DO), the final dataset included 1,049
[$NO_3^-$] observations from 162 wells sampled between 1993 and 2013 (Figure S1A). Additional details of the data selection, sources,
and manipulations may be found in the Supplemental Material.
Predictors were divided into two categories: static and dynamic (Table 1). Static predictors are those that either do not
change over the period of record, or annual records were limited. DO, for example, could potentially experience slight annual
variations, but data were not available to assign each nitrate sample a unique DO value. Instead, observations for each well were
assigned the average DO value observed from the well. This approximation was considered reasonable because nitrate isotopic
composition and DO data collected in the 1990s and by Wells et al. (2018) did not indicate any major changes to biogeochemical
processes over nearly two decades. Total travel time (from ground surface to the point of sampling) was strictly considered a static
predictor in this study and was used to link the nitrate-sampling year to a dynamic predictor value.
Dynamic predictors were defined in this study as data that changed temporally over the study period. Therefore, each
annual median [$NO_3^-$] was assigned a lagged dynamic value to represent the difference between the time of a particular surface
activity (e.g., timing of a particular irrigation practice) and when groundwater sampling occurred. Dynamic predictors were
available from 1946 to 2013 and included annual precipitation, Interstate Canal discharge, area under center pivot sprinklers, and
area of planted corn (Fig. 3). Dynamic predictors were included to assess their ability to optimize Random Forest groundwater
modelling and determine an appropriate lag time. Lag times were based on the vertical travel distance through both the vadose and
saturated zones (see Section 2.4). Area of planted corn was included as a proxy for fertilizer data, which were unavailable prior to
1987. However, analysis suggests there has been a 17% reduction (comparing the means of 1987-1999 to 2000-2012) in fertilizer
application rates per planted hectare, while area of planted corn has increased 16% (comparing the means of 1983-1999 to 2000-
2016) in recent decades (Wells et al., 2018). This trend may be attributed to improved fertilizer management by agricultural
producers. There was a likely trade-off in using this proxy; we were able to extend the period of record back to 1946, allowing for
analysis of a wider range of lag times in the model, but might have sacrificed some accuracy in recent decades when nitrogen
management may have improved. Lastly, vadose and saturated-zone transport rates were assumed to be constant over time (Wells
et al., 2018).

## 2.4 Vadose and Saturated-zone Transport Rate Analysis

Ranges of vertical velocities (transport rates) through the Dutch Flats vadose zone and saturated zone were estimated
from $^3H/^3He$ age-dating derived recharge rates. The vertical velocities were determined from results published for samples collected
in 1998 (Böhlke et al., 2007, Verstraeten et al., 2001a) and 2016 (Wells et al., 2018) as
$$V = \frac{R}{\theta},$$      (1)
where $R$ is the upper and lower bound of recharge rates (m/yr), and $\theta$ is the mobile water content in the vadose zone or porosity in
the saturated zone. The $^3H/^3He$ data were used in this study solely for constraining the range of potential transport rates to evaluate
in the vadose and saturated zones, and as a base comparison to model results. The age-data, however, were not used by the model
itself when seeking to identify an optimum transport rate combination. Throughout the text, unsaturated (vadose)-zone vertical
transport rates will be abbreviated as $V_u$, while saturated-zone vertical transport rates will be $V_s$. In the vadose zone, $\theta$ was assigned
a constant value of 0.13, which was calibrated previously using a vertical transport model for the Dutch Flats area (Liao et al.,
2012). In the saturated zone, $\theta$ was assigned a constant value of 0.35, equal to the value assumed previously for recharge
calculations (Böhlke et al., 2007). Vadose and saturated-zone travel times ($\tau$) then were calculated using Equation 2:
$$\tau = \frac{z}{V},$$      (2)
where $\tau$ is either vadose zone ($\tau_u$) or saturated zone ($\tau_s$) travel time in years, and z is the vadose-zone thickness ($z_u$) or distance
from the water table to well mid-screen ($z_s$) in meters.
Though Equations 1 and 2 do not explicitly consider horizontal groundwater flow, they are approximately consistent with
the distribution of groundwater ages (travel times from recharge), which increase with depth below the water table. Whereas
groundwater ages commonly increase exponentially with depth in idealized surficial aquifers with relatively uniform thickness and
distributed recharge (Cook and Böhlke, 2000), our linear approximation is based on several local observations, including (1) the
linear approximation is similar to the exponential approximation in the upper parts of idealized aquifers, (2) linear age gradients
may be appropriate in idealized wedge-shaped flow systems, as in some segments of the aquifer section (Figure 2), (3) focused
recharge under irrigation canals and distribution channels can cause distortion of vertical groundwater age gradients in
downgradient parts of the flow system, and (4) roughly linear age gradients were obtained from groundwater dating in the region,
though with substantial local variability (Böhlke et al., 2007). Discrete transport rates and travel times calculated from Equations
1 and 2 should be considered "apparent" rates and travel times, similar to apparent groundwater ages, which are based on imperfect
tracers and may be affected by dispersion and mixing. Nonetheless, the saturated open intervals of the monitoring wells used for
this study (< 6.1 m for shallow wells; 1.5 m for intermediate and deep wells) generally were short compared with the aquifer
thickness, such that age distributions within individual samples were relatively restricted in comparison to those of the whole
aquifer or of wells with long screened intervals. In addition, it is emphasized that the assumed mobile water content of 0.13 is a
calibrated parameter derived previously through inverse modelling and, as suggested by Liao et al. (2012), may have large
uncertainties due to the varying site-specific characteristics known to exist from one well to the next.
Because of the influence of canal leakage on both intermediate and deep wells (Böhlke et al., 2007), only recharge rates
from shallow wells were used to estimate initial values and permissible ranges of vadose-zone travel times. The mean ($\bar{x} = 0.38$
m/yr) and standard deviation ($\sigma = \pm0.23$ m/yr) of all the 1998 (n = 7) and 2016 (n = 2) shallow recharge rates were calculated.
Using $\bar{x} \pm 1\sigma$, a range of recharge rates from 0.15 to 0.61 m/yr (i.e., rates that varied by a factor of four) were converted to transport
rates ($V_u$) using Equation 1. Calculated transport rates resulted in 1.15 to 4.69 m/yr as the range of vadose-zone transport rates.
Expanding the upper and lower bounds, a minimum vadose-zone transport rate of 1.0 m/yr and maximum of 4.75 m/yr was applied.
Vertical transport rates in the vadose zone were increased by increments of 0.25 m/yr from 1.0 to 4.75 m/yr, resulting in 16 possible
vadose-zone transport rates to evaluate in the Random Forest model.
Mean ($\bar{x} = 0.84$ m/yr) and standard deviation ($\sigma = \pm0.73$ m/yr) of all shallow, intermediate, and deep well recharge rates
were included in identifying a range of saturated-zone recharge rates from 0.10 to 1.57 m/yr. A total of 35 and 8 recharge rates
were used from the Böhlke et al. (2007) and Wells et al. (2018) studies, respectively. Equation 1 was used to calculate saturated-
zone transport rates ($V_s$) of 0.28 and 4.49 m/yr. Saturated-zone transport rates were increased by increments of 0.25 m/yr, from
0.25 to 4.5 m/yr, resulting in 18 unique saturated-zone transport rates to evaluate in the Random Forest model. The range of
transport rates suggested by groundwater age-dating was large (more than an order of magnitude) and are considered to include
rates likely to be expected in a variety of field settings. Presumably, similar model constraints and results could have been obtained
without the prior age data and with some relatively conservative estimates.
Travel times $\tau_u$ and $\tau_s$ were calculated for each well based on $z_u$ and $z_s$, respectively. For every possible combination of
vadose and saturated-zone transport rates, a unique total travel time, $\tau_t$, was calculated for each well based on the vadose and
saturated-zone dimensions of that particular well.
$\tau_t = \tau_u + \tau_s,$                                                     (3)
The total travel times from Equation 3 were used to lag dynamic predictors relative to each nitrate sample date. For
instance, a nitrate sample collected in 2010 at a well with a 20-year total travel time (e.g., $\tau_u = 10$ yrs and $\tau_s = 10$ yrs) would be
assigned the 1990 values for precipitation (450 mm), Interstate Canal discharge (0.4 km$^3$/yr), center pivot irrigated area (2484
hectares), and area of planted corn (8905 hectares).
A total of 288 unique transport rate combinations (corresponding to different combinations of the 16 vadose and 18
saturated-zone transport rates) were evaluated. Each transport rate combination incorporated up to 1,049 groundwater [NO$_3^-$] values
in the Random Forest model.

**2.5 Random Forest Application**

Random Forests are created by combining hundreds of unskilled regression trees into one model ensemble, or "forest",
which collectively produce skilled and robust predictions (Breiman, 2001). Models of groundwater [NO$_3^-$] were developed using
five-fold cross validation (Hastie et al., 2009), where each fold was used to build the model (training data) four times, and held out
once (testing data). The maximum and minimum of the groundwater [NO$_3^-$] and each predictor were determined and placed into
each fold for training models to eliminate the potential for extrapolation during validation. The four folds designated to build the
model also underwent a nested five-fold cross validation, as specified in the *trainControl* function within the *caret* (Classification
and Regression Training) R package (Kuhn, 2008; R Core Team, 2017). Functions in *caret* were used to train the Random Forest
models. We repeated the five-fold cross validation process five times to create a total of 25 models, similar to the approach used
by Nelson et al. (2018), in order to assess sensitivity of model performance to the data assigned to the training and testing folds.
Permutation importance, partial dependence and Nash-Sutcliffe Efficiency (NSE)) were quantified to evaluate model
performance and to interpret results. NSE (Nash and Sutcliffe, 1970) was calculated as
$NSE = 1 - \left[ \dfrac{\sum_{i=1}^{n}\left(Y_i^{obs} - Y_i^{pred}\right)^2}{\sum_{i=1}^{n}\left(Y_i^{obs} - Y^{mean}\right)^2} \right],$    (4)
where $n$ is the number of observations, $Y_i^{obs}$ is the i[th] observation of the response variable ([NO$_3^-$]), $Y_i^{pred}$ is the i[th] prediction from
the Random Forest model, and $Y^{mean}$ is the mean of observations $i$ through $n$. Values from negative infinity to 0 suggest the mean
of the observed [NO$_3^-$] would serve as a better predictor than the model. When NSE = 0, model predictions are as accurate as that
of a model with only the mean observed [NO$_3^-$] as a predictor. From 0, larger NSE values indicate a model's predictive ability
improves, until NSE = 1, where observations and predictions are equal. NSE was calculated for both the training and testing data.
For each tree, a random bootstrapped sample (i.e., data randomly pulled from the dataset, sampled with replacement) is
extracted from the dataset (Efron, 1979), as well as a random subset of predictors to consider fitting at each split. Thus, each tree
is grown from a bootstrap sample and random subset of predictors, making trees random and grown independent of the others.
Observations not used as bootstrap samples are termed out-of-bag (OOB) data.
When building a tree, all [NO$_3^-$] from the bootstrap sample are categorized into terminal nodes, such that each node is
averaged and yields a predicted [NO$_3^-$]. The performance and mean squared error (MSE) of a Random Forest model is evaluated
by comparing the observed [NO$_3^-$] of the OOB data to the average predicted [NO$_3^-$] from the forest. OOB data from the training
dataset may be used to evaluate both permutation importance, referred to in the rest of this text as variable importance, and partial
dependence. Variable importance uses percent increase in mean squared error (%$_{inc}$MSE) to describe predictive power of each
predictor in the model (Jones and Linder, 2015). During this process, a single predictor is permuted, or shuffled, in the dataset.
Therefore, each observed [NO$_3^-$] has the same relationship between itself and all predictors, except one permuted variable. The
%$_{inc}$MSE of a variable is determined by comparing the permuted OOB MSE to unpermuted OOB MSE. Important predictors will
result in a large %$_{inc}$MSE, while a variable of minor importance does little to impact a model's performance, as suggested by a low
%$_{inc}$MSE value.
Partial dependence curves serve as a graphical representation of the relationship between [NO$_3^-$] and predictors in the
Random Forest model ensemble (Hastie et al., 2009). In these models, the y-axis of a partial dependence plot represents the average
of the OOB predicted [NO$_3^-$] at a specific x-value of each predictor.
**3 Results and Discussion**
This study addressed a relatively unexplored use of Random Forest, which was to identify optimal lag times based on
testing a range of transport rate combinations through the vadose and saturated zones, historical [NO$_3^-$], and the use of easily
accessible environmental datasets.
**3.1. Relative Importance of Transport Time and Dynamic Variables**
In our initial modelling (using both static and dynamic predictors), we anticipated that we could use the Random Forest
model with the highest NSE to identify the optimal pair of vadose and saturated-zone transport rates. However, no clear pattern
emerged among the different models (Fig. 4). Given the small differences and lack of defined pattern in testing NSE values, we
selected ten transport rate combinations (the five top performing models, plus four transport rate combinations of high and low
transport rates, and one intermediate transport rate combination) for further evaluation of variable importance and sensitivity to a
range of transport rate combinations (Table 2). Median total travel time ranked third in variable importance, while the four dynamic

variables consistently had the four lowest rankings (Fig. 5). Total travel time also had the greatest variability in importance among the fifteen variables, with a range of 18.4% between the upper and lower values, suggesting some model sensitivity to lag times. Excluding total travel time, the remaining variables had an average variable importance range of 6%.

Dynamic variables had little influence on the model, despite common potential linkages to groundwater [$NO_3^-$] (Böhlke et al., 2007; Exner et al., 2010; Spalding et al., 2001). A pattern emerged among dynamic variables where the stronger the historical trend of the predictor, the greater the importance of the predictor (Fig. 3; Fig. 5). For instance, center pivot irrigated area (highest ranking dynamic variable) had the least noise and the most pronounced trend, while annual precipitation (lowest ranking variable) was highly variable and lacked any trend over time (Fig. 3), and also may not be a substantial source of recharge (Böhlke et al., 2007). Further exploration could be done to test more refined and/or spatially varying predictors – for instance, annual median rainfall intensity for the growing season might have a more direct connection to nitrate leaching than total annual precipitation. However, rainfall intensity data are not readily available. Likewise, availability of a long-term, detailed fertilizer loading dataset would be advantageous in providing a more substantiated conclusion regarding the viability of applying dynamic variables to determine vadose and saturated-zone lag. Dynamic variables could be of more use in other study areas that undergo relatively rapid and pronounced changes (e.g., land use). In future work, the model sensitivity to dynamic variables could be tested through formal sensitivity analysis and/or automated variable selection algorithms (Eibe et al., 2016).

Ultimately, results from initial analyses suggest that (1) the dynamic data did little to improve model performance, and (2) Random Forest was not able to relate the four considered dynamic predictors to [$NO_3^-$] in a meaningful way that could be used to estimate lag time It is likely the influence of these dynamic predictors is dampened as nitrate is transported from the surface to wells such that data-driven approaches are unable to sort through noise to identify relationships.

**3.2 Use of Random Forest to determine transport rates**

Due to their low relative importance as predictors, all four dynamic predictors were removed in the subsequent analysis. As discussed above, a notable variation in total travel time %$_{inc}$MSE was observed in Fig. 5, suggesting model sensitivity to this variable. Additionally, a relationship between travel time and [$NO_3^-$] has been suggested in the Dutch Flats area through previous studies (Böhlke et al., 2007; Wells et al., 2018). Therefore, a second analysis of just the 11 static predictors was performed over the full range of vadose and saturated transport rates (i.e., 288 combinations). However, in the second analysis, model sensitivity to total travel time – evaluated with respect to the transport rate combination corresponding to the largest %$_{inc}$MSE of total travel time – was used to determine a distinguished transport rate combination. In other words, models were re-trained and tested for all transport rate combinations, each of which produced a unique set of values for the total travel time variable. As described in Section 2.5, the %$_{inc}$MSE value for total travel time was then based on the error induced in the model by permuting the calculated total travel times across all the nitrate observations (i.e., randomly shuffling the total travel time variable, and thus disturbing the structure of the dataset).

The Random Forest models were useful in identifying the relative magnitudes of $V_u$ and $V_s$ that led to high %$_{inc}$MSE. Based on the heat map of %$_{inc}$MSE, a band of transport rate combinations with consistently high %$_{inc}$MSE was visually apparent (Fig. 6). The upper and lower bounds of the band translate to transport rate ratios ($V_s/V_u$) ranging from 0.9 to 1.5, and are values that could be useful in constraining recharge and/or transport rate estimates in more complex mechanistic models of the Dutch Flats area, as part of a hybrid modelling approach. This is especially important because recharge is one of the most sensitive parameters in a groundwater model (Mittelstet et al., 2011), yet one with high uncertainty. Whereas a saturated-zone velocity that is greater than a vadose-zone velocity would be unexpected in many unconsolidated surficial aquifers receiving distributed recharge, the statistical machine learning results are consistent with two contrasting primary recharge processes in the Dutch Flats

area: (1) diffuse recharge from irrigation and precipitation across the landscape, and (2) focused recharge from leaking irrigation conveyance canals.

The $\%_{inc}$MSE of total travel time in the second analysis (using only static variables) ranged from 20.6 to 31.5%, with the largest $\%_{inc}$MSE associated with vadose and saturated-zone transport rates of 3.50 m/yr and 3.75 m/yr, respectively (Fig. 6), and the top four predictors for this transport rate combination were total travel time, vadose-zone thickness, dissolved oxygen, and saturated thickness (Fig. 7). Converting those vadose and saturated-zone transport rates to recharge rates yielded values of 0.46 m/yr and 1.31 m/yr, respectively. Such a large difference between the two recharge values is consistent with the hydrologic conceptual model of the Dutch Flats area. In fact, both model recharge rates compare favourably with recharge rates calculated from the previous Dutch Flats studies using $^3H/^3He$ age-dating (Böhlke et al., 2007; Wells et al., 2018). For instance, the recharge rate determined from the vadose-zone transport rate in this study (0.46 m/yr) was comparable to the mean recharge rate of 0.38 m/yr (n = 9) from groundwater age-dating at shallow wells, which are most representative of diffuse recharge below crop fields that are present across most of the study area (e.g., Figure S2). Additionally, the recharge rate (1.31 m/yr) determined from the saturated-zone transport rate was consistent with the mean recharge value derived from groundwater ages in intermediate wells (1.22 m/yr, n = 13; Böhlke et al., 2007; Wells et al., 2018). Intermediate wells are variably impacted by focused recharge from canals in upgradient areas. Given the similarity in diffuse recharge and focused recharge estimates from both Random Forest and groundwater age-dating, the transport rate ratios (1.2 and 1.1, respectively) were consistent. That is, the Random Forest modelling framework produced transport rates consistent with the major hydrological processes in Dutch Flats both in direct (i.e., transport rate estimates) and relative (i.e., transport rate ratio) terms.

Assuming the Random Forest approach has accurately captured the two major recharge processes (diffuse recharge over crop fields and focused recharge from canals), a comparison of recharge rates from all sampled groundwater wells representative of recharge to the groundwater system as a whole (0.84 m/yr, n = 43) to the recharge rates from Random Forest modelling (0.46 and 1.31 m/yr) would provide an estimate of the relative importance of diffuse versus focused recharge on overall recharge in Dutch Flats. Under these assumptions, diffuse recharge would account for approximately 55%, while focused recharge would account for about 45% of total recharge in the Dutch Flats area. Similarly, Böhlke et al. (2007) concluded that these two recharge sources contributed roughly equally to the aquifer on the basis of groundwater age profiles, as well as from dissolved atmospheric gas data indicating mean recharge temperatures between those expected of diffuse infiltration and focused canal leakage.

Partial dependence plots, which illustrate the impact a single predictor has on $[NO_3^-]$ in the model with respect to other predictors (Fig. 8), largely reflect the conceptual understanding of the system from previous studies including Böhlke et al. (2007) and Wells et al. (2018). Key features that strengthen confidence in the Random Forest modelling include (1) depth to bottom screen, where groundwater $[NO_3^-]$ is lower at greater depths, (2) the effects of minor and major canals, where groundwater $[NO_3^-]$ in the vicinity of canals is diluted by canal leakage, and the influence of major canals extends a longer distance when compared to that of minor canals, (3) land surface elevation, where elevations indicating proximity to major canals are associated with relatively lower groundwater $[NO_3^-]$, and (4) DO concentration, where higher DO concentration is linked to higher groundwater $[NO_3^-]$. We note that decreasing DO and $[NO_3^-]$ with groundwater age can be explained by DO reduction and historical changes in $[NO_3^-]$ recharge, whereas groundwater chemistry and nitrate isotopic data recorded in both this study and previous Dutch Flats studies suggest denitrification was not a major factor in this alluvial aquifer.

The partial dependence plot (Fig. 8) for total travel time exhibits a pronounced threshold, where $[NO_3^-]$ is markedly higher for groundwater with travel time less than seven years. It is possible this reflects long-term stratification of groundwater $[NO_3^-]$, stemming from the suggested patterns stated above as nitrate varies with aquifer depth due to the influences of diffuse and focused recharge in the region. This seven-year threshold is slightly lower than a previous estimate of mean groundwater age in the aquifer

(8.8 years; Böhlke et al., 2007; where groundwater age excludes vadose-zone travel time) and suggests that shallow groundwater
can respond relatively rapidly to changes in nitrogen management in the Dutch Flats area.
**3.3 Opportunities and limitations of Random Forest approach in estimating lag times**

Overall, results suggest that  in a complex system such as Dutch Flats, Random Forest was able to identify reasonable
transport rates for both the vadose and saturated zones, and with additional validation, this method may offer an inexpensive (i.e.,
compared to groundwater age-dating across a large monitoring well network and/or complex modelling) and reasonable technique
for estimating lag time from historical monitoring data. Further, this approach allows for additional insight on groundwater
dynamics to be extracted from existing monitoring data. However, this study was conducted in the context of a larger project
(Wells et al., 2018) and built on prior research on groundwater flow and [$NO_3^-$] in the study area (Böhlke et al., 2007). Therefore,
it is critical in future work to incorporate site-specific knowledge, process understanding, and approaches for increasing
interpretability of machine learning models (Lundberg et al., 2020, Saia et al., 2020), as highlighted in key considerations below.
Some key considerations for future application of this approach include:
(1)  The Random Forest approach might be useful for estimating future recharge and [$NO_3^-$] using multiple potential
management scenarios, as long as considered management scenarios fall within the range of historical observations used
to train the model. This information could be used to inform policy makers of the impact that current and future
management decisions will have on recharge and [$NO_3^-$].
(2)  The Dutch Flats overlies a predominantly oxic aquifer, where nitrate transport is mostly conservative. In aquifers with
heterogeneity in denitrification potential and/or distinct nitrate extinction depths (Liao et al., 2012; Welch et al., 2011),
this approach may be biased toward oxic portions of the aquifer where the nitrate signal is preserved. Similarly, vertical
profiles of [$NO_3^-$] and isotopic composition in the vadose zone could provide valuable data to investigate (1) the amount
of nitrate stored in the vadose zone, and (2) whether nitrate undergoes any biogeochemical changes while being
transported through the vadose zone to the water table.
(3)  While estimates of vadose and saturated-zone transport rates determined from %$_{inc}$MSE are consistent with previous
studies, the predictive performance of the selected model (based on NSE and visual inspection of predicted versus
observed nitrate plots) was not substantially different than other models tested. In other words, the "optimal model" was
only weakly preferred in terms of predicting [$NO_3^-$]. Testing the approach of using %$_{inc}$MSE in other vadose and saturated
zones, with substantial comparison to previous transport rate estimates, is warranted. This would be especially valuable
in an area with a well-defined input function for nitrate that could be compared to a reconstructed input function from the
model. Further, in aquifer settings with relatively evenly distributed recharge, optimized travel times to wells could be
used to estimate the infiltration date of samples, thus providing an optimized view of historical variation of [$NO_3^-$] entering
the subsurface, as illustrated in Figure S1B. In the Dutch Flats area, however, such an analysis is complicated by effects
of subsurface nitrate dilution by local recharge from canal leakage.
(4)  Despite potential non-uniqueness in prediction metrics, the heat map of %$_{inc}$MSE did reveal an orderly pattern suggesting
consistent transport rate ratios. For modelling efforts where recharge rates are a key calibration parameter, identification
of a range of reasonable recharge rates, and/or the ratio of recharge rates from diffuse and focused recharge sources for a
complex system will reduce model uncertainty and improve results. This statistical machine learning approach, which
essentially leverages nitrate as a tracer (albeit with an unknown input function in this case), may provide valuable insight

to complement relatively expensive groundwater age-dating or vadose-zone monitoring data, or perhaps as a standalone

approach for first-order approximations.

(5) The demonstrated statistical machine learning approach is apparently well-suited for drawing out transport rate

information from a site with two distinct recharge sources (diffuse versus focused recharge sources) driving the

groundwater nitrate dynamics. Further testing is needed at sites where recharge and nitrate dynamics are more subtle.

## 4 Conclusions

The Dutch Flats area exhibits large variations in $[NO_3^-]$ throughout a relatively small region in western Nebraska. Long-
term groundwater $[NO_3^-]$ monitoring and previous groundwater age-dating studies in Dutch Flats provided an opportune setting to
test a new application of statistical machine learning (Random Forest) for determining vadose and saturated-zone transport rates.
Overall results suggest Random Forest has the capability to both identify reasonable transport rates (and lag time) and key variables
influencing groundwater $[NO_3^-]$, albeit with potential for non-unique results. Limitations were also identified when using dynamic
predictors to model groundwater $[NO_3^-]$. Utilizing only static predictors, and Random Forest's ability to evaluate variable
importance, vadose-zone and saturated-zone transport rates were selected based on model sensitivity to changing the total travel
time predictor. In other words, total travel time variable importance was evaluated for 288 different transport rate combinations,
and the combination with a total travel time having the largest influence over the model's ability to predict $[NO_3^-]$ was selected for
additional examination. This analysis identified a vadose-zone and saturated-zone transport rate combination consistent with rates
previously estimated from $^3H/^3He$ age-dating in Böhlke et al. (2007) and Wells et al. (2018), indicating a combination of distributed
and focused sources of irrigation recharge to this aquifer
Future studies could include assessments of the proper conditions for application of dynamic predictors and include
comparisons of data-driven analyses with complementary datasets and/or modelling (e.g., field-based recharge rate estimates,
finite-difference flow model). Despite noted limitations, partial dependence plots and relative importance of predictors were largely
consistent with previous findings and mechanistic understanding of the study area, giving greater confidence in model outputs.
The influence of canal leakage on groundwater recharge rates and $[NO_3^-]$, for example, was consistent with previous Dutch Flats
studies. Partial dependence plots suggest a threshold of higher $[NO_3^-]$ for groundwater with total travel time (vadose and saturated-
zone travel times, combined) of less than seven years, indicating the potential for relatively rapid groundwater $[NO_3^-]$ response to
widespread implementation of best management practices. Additionally, research is needed to determine the minimum number of
observations needed to effectively apply the framework shown here.

**Author contribution:** TG, AM, and NN were responsible for conceptualization. MW and NN developed the model code and MW
performed formal analysis. MW prepared the manuscript from his M.S. thesis with contributions from all co-authors, including
JKB. TG was responsible for project administration and funding acquisition.

**Acknowledgements:** The authors acknowledge the North Platte Natural Resources District for providing technical assistance and
resources, including long-term groundwater nitrate data accessed via the Quality-Assessed Agrichemical Contaminant Database
for Nebraska Groundwater. We thank Steve Sibray and Mason Johnson for their support in field sampling efforts and Les Howard
for cartography. Models were run on the Holland Computing Center (HCC) cluster at the University of Nebraska-Lincoln. We also
thank Christopher Green, Sophie Ehrhardt, Pia Ebeling, and two anonymous reviewers for helpful comments on earlier versions
of the paper. Any use of trade, firm, or product names is for descriptive purposes only and does not imply endorsement by the U.S.
Government.

**Funding:** This work was supported by the U.S. Geological Survey 104b Program (Project 2016NE286B), U.S. Department of
Agriculture—National Institute of Food and Agriculture NEB-21-177 (Hatch Project 1015698), and Daugherty Water for Food
Global Institute Graduate Student Fellowship.

**Supplemental Material:** An online file accompanying this article contains additional figures, tables, and details of methods used
for the study.

**Code and Data Availability:** Code is available on request. Data used in the random forest model and described in the supplemental
material is available via the University of Nebraska – Lincoln Data Repository (https://doi.org/10.32873/unl.dr.20200428).

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

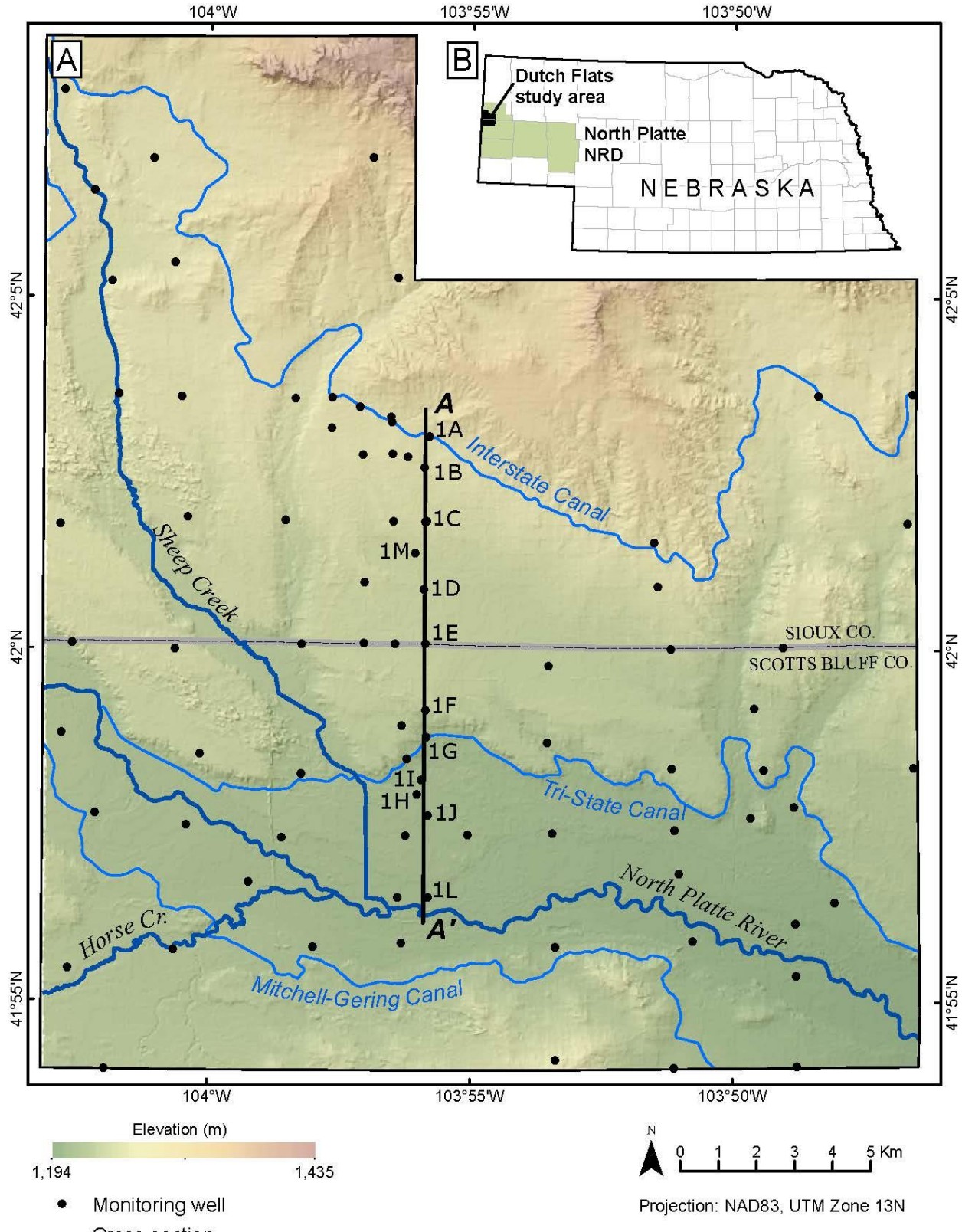

**Figure 1: Dutch Flats study area (A) overlain by 30 m Digital Elevation Model (USGS, 1997). The study area is located within the North**
**Platte Natural Resources District of western Nebraska (B). Depending on data availability, multiple wells (well nest) or a single well may**
**be found at each monitoring well location. Transect A-A' represents the location and wells displayed in the Fig. 2 hydrogeologic cross-**
**section.**


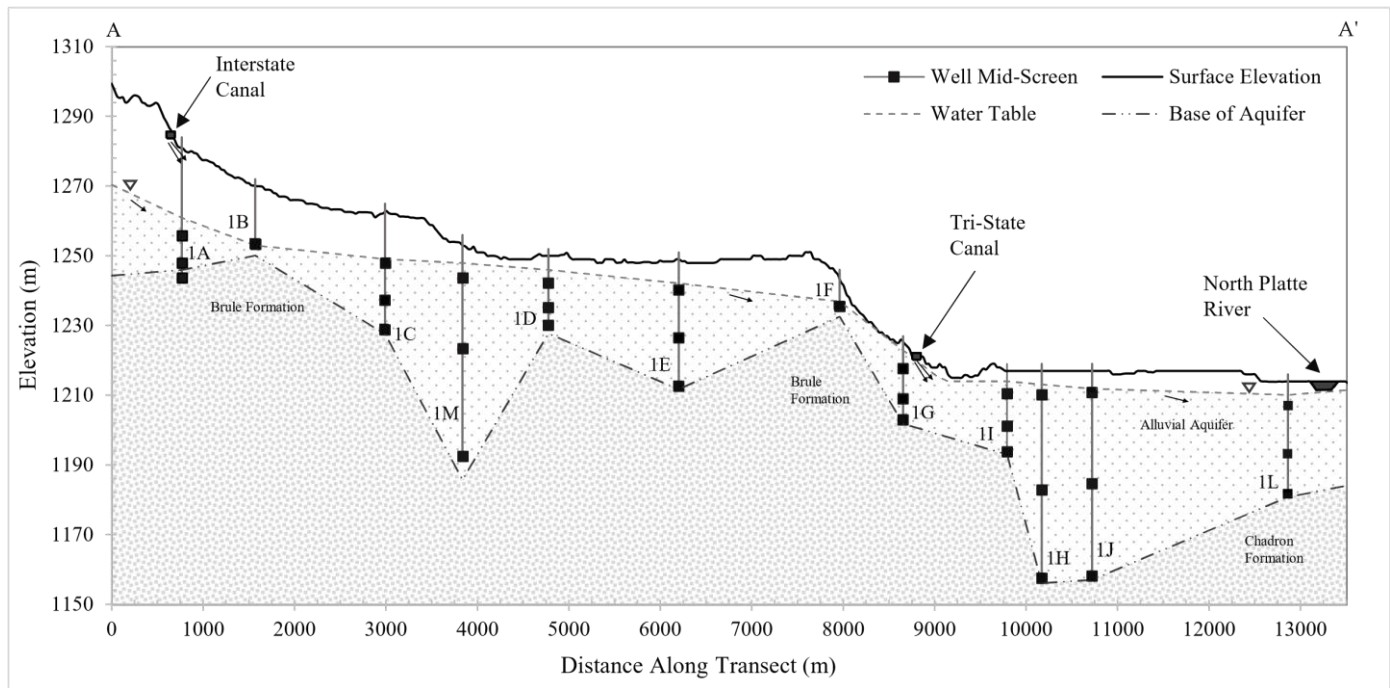

**Figure 2: Cross-section along representative well transect (see Fig. 1) within the Dutch Flats area. Surface elevation data were derived from a 30-meter Digital Elevation Model (USGS, 1997). Water surface and base of aquifer elevations were sourced from a 1998 Dutch Flats study (Böhlke et al., 2007, Verstraeten et al., 2001a, 2001b). Small black arrows beneath the surface indicate general groundwater flow direction.**

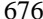

**Figure 3: Time series plots of all four dynamic predictors. Figures represent (a) annual precipitation, (b) Interstate canal discharge, (c) center pivot irrigated area, and (d) area of planted corn from 1946 to 2013.**


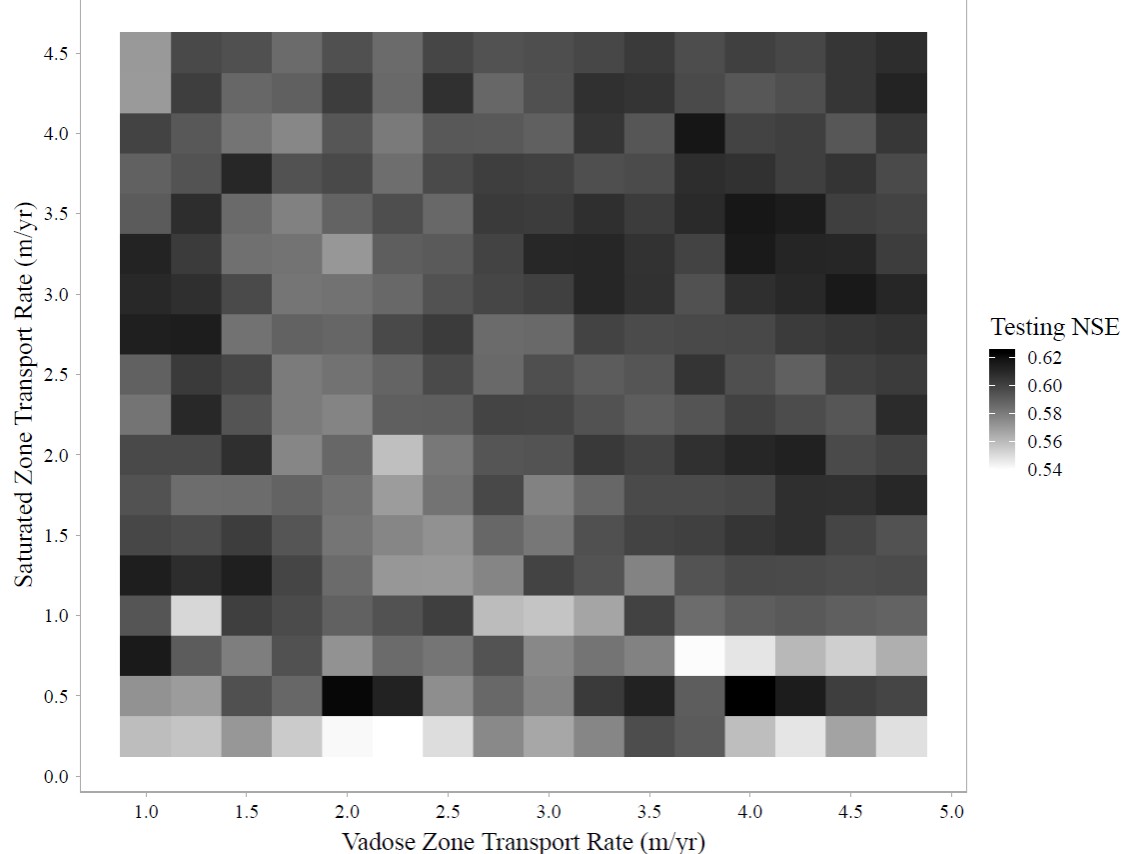

**Figure 4: Heat map of testing NSE results from 288 vadose and saturated-zone transport rate combinations. Testing NSE in this figure**
**is the median of all 25 model outputs from each of the 288 transport rate combinations. No clear pattern of optimal vadose and saturated-**
**zone transport rate combinations was observed.**

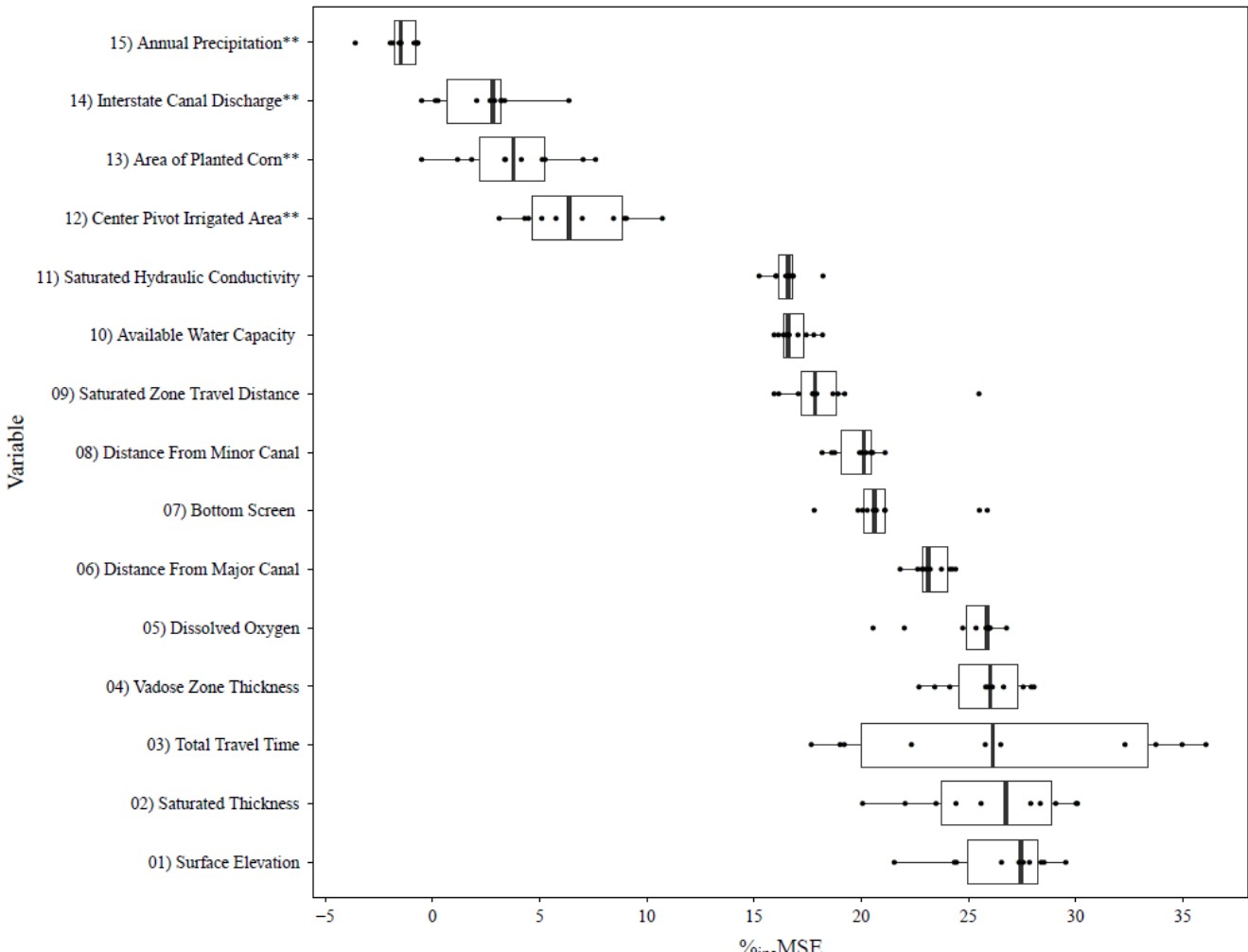

Figure 5: Boxplot of the %$_{inc}$MSE from the ten transport rate combinations shown in Table 2. Each boxplot has ten points for each transport rate combination, representing the median %$_{inc}$MSE from the 25 models (five-fold cross validation, repeated 5 times). A larger %$_{inc}$MSE suggests the variable had a greater influence on a model's ability to predict [NO$_s^-$]. **Denotes dynamic predictors.

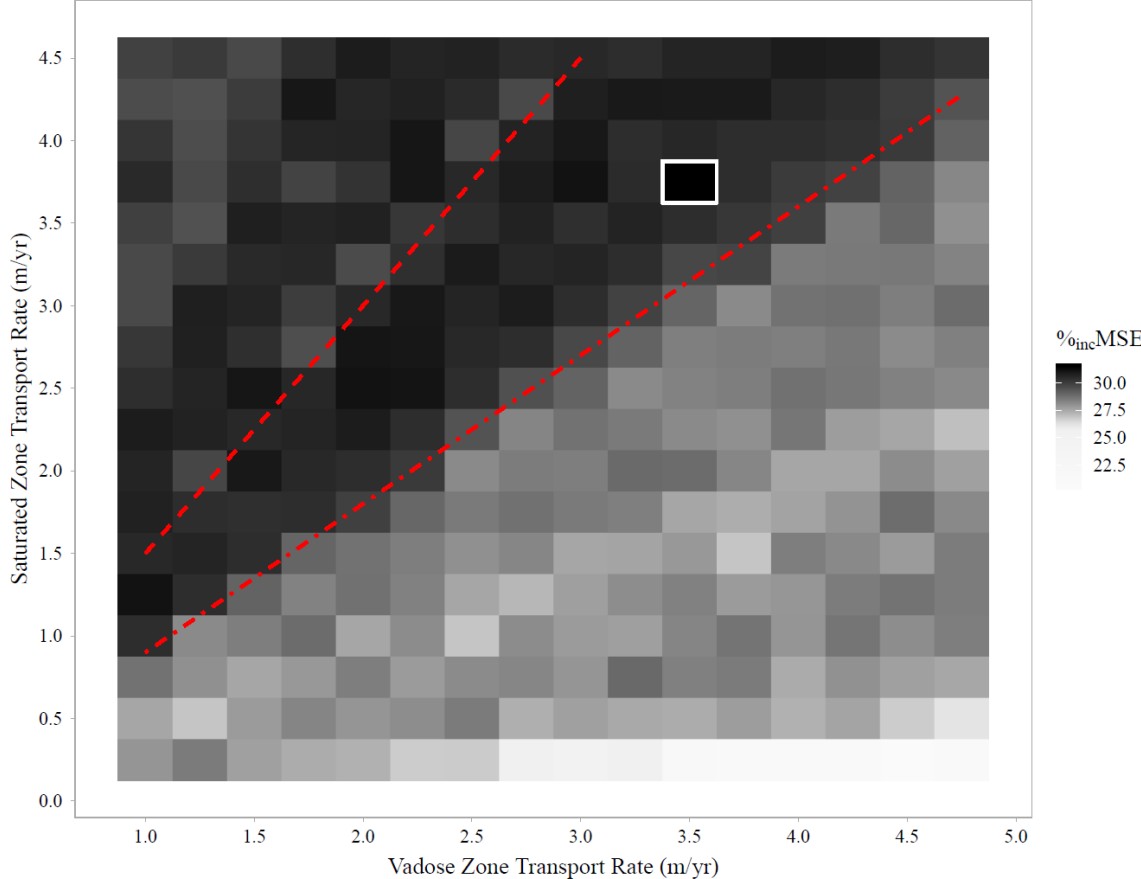

**Figure 6: Heat map of %$_{inc}$MSE (median from 25 models) from variable importance of total travel time for each of the 288 transport rate combinations evaluated. Red dashed lines indicate upper ($V_s$ / $V_u$ = 1.5, long dashes) and lower (0.9, short dashes) bounds of the band of transport rate combinations with consistently higher %$_{inc}$MSE. The white square highlights the single transport rate combination with the highest %$_{inc}$MSE.**

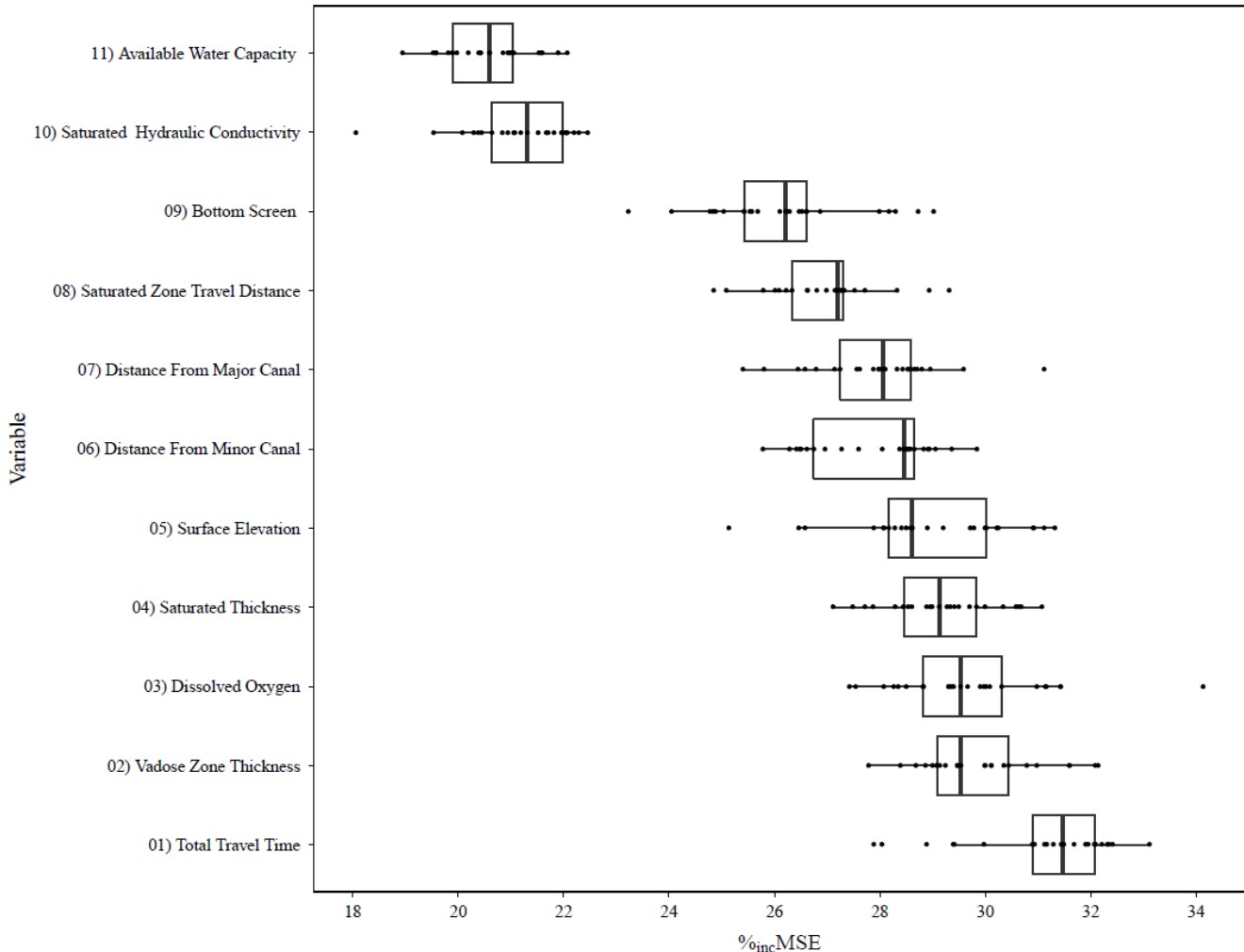

**Figure 7: Plot from secondary analysis exploring variable importance of the transport rate combination with the largest median**
**$\%_{inc}$MSE in total travel time ($V_u$ = 3.5 m/yr; $V_s$= 3.75 m/yr). Each point is from one of 25 Random Forest models run for this evaluation.**
**A larger $\%_{inc}$MSE suggests the variable had a greater influence on a model's ability to predict [NO$_s^-$].**


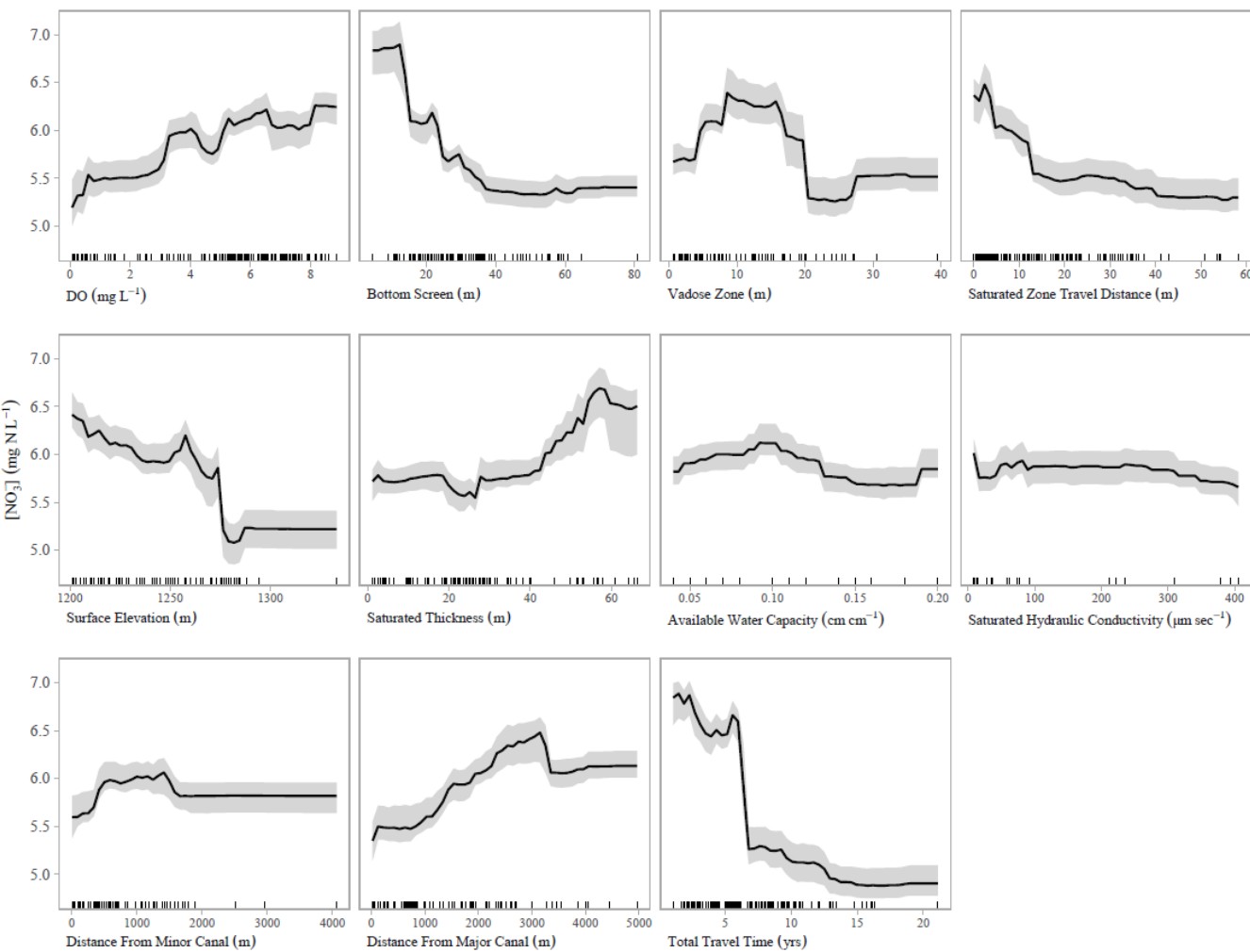

Figure 8: Partial dependence plot for model evaluating transport rate combination of $V_u$ = 3.5 m/yr and $V_s$= 3.75 m/yr. Tick marks on each plot represent predictor observations used to train models.

**Table 1. List of the 15 predictors used for Random Forest evaluation. Average (avg.) and median (med.) values are shown.**

| Predictor | Units | Predictor Type | Source |
|---|---|---|---|
| Center Pivot Irrigated Area (avg. = 2618; med. = 1037)[a] | hectare | Dynamic | NAIP; NAPP; Landsat-1,5, 7, 8[b] |
| Interstate Canal Discharge (avg. = 0.53; med. = 0.55)[a] | $km^3 \, yr^{-1}$ | Dynamic | USBR (2018) |
| Area of Planted Corn (avg. = 8065; med. = 7869)[a] | hectare | Dynamic | NASS (2018) |
| Precipitation (avg. = 384; med. = 377)[a] | $mm \, yr^{-1}$ | Dynamic | NOAA (2017) |
| Available Water Capacity (avg. = 0.1; med. = 0.1) | $cm \, cm^{-1}$ | Static | NRCS (2018) |
| Dissolved Oxygen (avg. = 4.6; med. = 5.4) | $mg \, L^{-1}$ | Static | C. Hudson, Personal Communication (2018) |
| Distance from a Major Canal (avg. = 1462.2; med. = 1161.4) | m | Static | USGS (2012)[b] |
| Distance from a Minor Canal (avg. = 633.2; med. = 397.6) | m | Static | USGS (2012)[b] |
| Bottom Screen (avg. = 26.9; med. = 24.4) | m | Static | UNL (2016)[b] |
| Saturated Hydraulic Conductivity (avg. = 68; med. = 28) | $\mu m \, sec^{-1}$ | Static | NRCS (2018) |
| Saturated Thickness (avg. = 30.2; med. = 27.6) | m | Static | T. Preston, Personal Communication (2017)[b] |
| Saturated-Zone Travel Distance (avg. = 13.3; med. = 7) | m | Static | UNL (2016)[b] |
| Surface Elevation (DEM) (avg. = 1244; med. = 1248) | m | Static | USGS (1997) |
| Total Travel Time (avg. = 6.4; med. = 5.7)[c] | years | Static | UNL (2016)[b] |
| Vadose-Zone Thickness (avg. = 9.9; med. = 7.3) | m | Static | T. Preston, Personal Communication (2017); A. Young, Personal Communication (2016) |

[a] Average and median span from 1946 to 2013

[b] Data required further analysis to yield calculated values; data sources are USDA (2017) and USGS (2017)

[c] Average and Median reflects transport rates of $V_u$ = 3.5 m/yr and $V_u$ = 3.75 m/yr

**Table 2. Summary of ten vadose and saturated-zone transport rate combinations selected from 288 unique potential combinations from the analysis including dynamic variables.**

| | Vadose-zone Transport Rate (m/yr) | Sat. Zone Transport Rate (m/yr) | Test NSE | [NO$_3^-$] Observations[a] | Total Travel Time (yrs) Mean (±1σ) | Total Travel Time (yrs) Median |
|---|---|---|---|---|---|---|
| Five Top-Performing Transport Rates | 4.00 | 0.50 | 0.623 | 878 | 19.9 (± 15.8) | 11.3 |
| | 2.00 | 0.50 | 0.622 | 861 | 21.6 (± 15.0) | 16.5 |
| | 3.75 | 4.00 | 0.617 | 1049 | 6 (± 3.7) | 5.4 |
| | 4.00 | 3.50 | 0.617 | 1049 | 6.3 (±4.1) | 5.7 |
| | 4.50 | 3.00 | 0.616 | 1049 | 6.7 (± 4.7) | 5.7 |
| Extreme and Midrange Transport Combinations | 4.75 | 4.50 | 0.608 | 1049 | 5.1 (± 3.2) | 4.6 |
| | 2.75 | 2.25 | 0.599 | 1049 | 9.6 (± 6.3) | 8.5 |
| | 1.00 | 4.50 | 0.570 | 1049 | 12.6 (± 7.7) | 10.8 |
| | 1.00 | 0.25 | 0.559 | 607 | 26.7 (± 13.3) | 20.6 |
| | 4.75 | 0.25 | 0.548 | 664 | 21.3 (± 15.0) | 14.9 |

[a] In cases with slow transport rates, lag times were relatively long and not all [NO$_3^-$] data could be used in the model. For example, a slow transport rate combination resulting in a lag time with the infiltration year prior to 1946 could not be included. Thus, some models were ultimately based on <1,049 observations.