# Peer review of "term groundwater monitoring data and statistical machine learning"

_Hydrology and Earth System Sciences, 2020_

## Referee Comment (RC1) · Anonymous Referee #1 · 5 Jun 2020

The paper by Wells et al. with title: "Determination of vadose and saturated-zone nitrate lag times using long-2 term groundwater monitoring data and statistical machine learning" provides an interesting study regarding the application of Random Forest Statistical method using data to estimate vadose and saturated-zone vertical velocities (transport rates) for the determination of subsurface lag times. The authors use Random Forest Regression to leverage existing long-term groundwater nitrate concentration (referred to as [NO3-] hereafter). The study area is located in the western Nebraska counties in which the crop area is increased over the years. Hence, regulators and stakeholders in agricultural landscapes are increasingly in need of more precise and local lag time information to better evaluate and apply regulations and best management practices for the reduction of groundwater nitrate concentrations. The authors use R programming packages adding 15 predictors to estimate the vadose (unsaturated zone) and groundwater (saturated zone) transport rates and lag times for an intensively monitored alluvial aquifer at Dutch Flats area. The manuscript is well written and the innovation of the study is clear. However, the geological/Hydrogeological information is missing a deeper analysis of the dynamic and static predictors as well. My recommendation is publication after major revisions.

General comments - Geology/Hydrogeology is missing. Provide a hydrogeological map, cross section, hydraulic characteristics of the aquifer etc. - Add more information about nitrate and its processes. - Fig1.Change the maps. The figure needs to be more attractive. Add coordinate system. - The literature is out of date. - Discuss the role of Nitrate isotopes for future contribution in this concept. Recent article provide the interaction between surface and groundwater bodies using nitrate isotopes which might be helpful in future works.

---

## Referee Comment (RC2) · Scott Gardner (Referee) · 20 Jun 2020

This paper utilizes a very interesting statistical methodology for evaluating nitrate transit times across saturated and unsaturated zones. General comments - The study presents the environmental setting well in terms of soil, climate, and land use, however, more specific information (cross-sections or maps) on the geologic setting would be useful in evaluating spatial variability in lag times. - The distance between the monitoring wells evaluated and the screens that are sampled to the sources of nitrate (probably fields) are not touched on in the manuscript and might be useful in explaining

variance in lag times. - Perhaps land use might also be important to consider nearby the wells, as interception, evapotranspiration, and other land use specific processes could be relevant to nitrate lag times.

Specific comments line 17 - I am not sure you need to include the part about it not being common to have unsaturated velocities slower than saturated, this has been the case in other studies and is not out of the ordinary (fractured bedrock aquifers, karst, etc.) line 79 - perhaps provide a reference explaining the importance of canals in the region for readers that are not familiar with the study area. line 107 - here and everywhere after it is not clear what is meant by screen length, is this the depth bgs that the screen begins, or the size of the screen? please clarify line 157 - what is meant by 'bootstrapped' readers which are unfamiliar with computer science jargon may have trouble with this please clarify. line 234 - what was the reasoning behind selecting 1 standard deviation for an acceptable range of results? If this selection was arbitrary then it should be made clear.

figure s1 please change the colours on the nitrate concentrations to better contrast the results

Conclusion The paper is well written and presents a great analysis using a interesting statistical modeling approach. The long term nature of the study data gives observations strong credence. Adding more specific spatial analyses between different well locations in the region might help explain the high variance in model results. My recommendation is acceptance after major revisions.

---

## Editor Comment (EC1) · Nunzio Romano (Editor) · 6 Jul 2020

Dear Authors: Two out of the three reviewers have posted some comments and raised interesting concerns. While waiting for other appraisals to come in, I'd suggest you should start providing some preliminary replies to the comments received so far. In this way, the discussion step of the journal becomes even more alive.

---

## Referee Comment (RC3) · Sophie Ehrhardt (Referee) · 14 Jul 2020

The new paper "Determination of vadose and saturated-zone nitrate lag times using long-term groundwater monitoring data and statistical machine learning" by Wells et al., presents an innovative approach to estimate vadose-zone and saturated-zone lag times using long-term groundwater nitrate data. The use of statistical machine learning could be an alternative to expensive groundwater age-dating techniques and has the computational power to uncover nonlinear trends. Both are convincing arguments for the application of the Random Forest analyses and provide valuable information for

groundwater management. General comments Clearly written, but with some missing further information (see specific comments). Nicely explained method section! Especially for natives in ML very instructive. Interesting analysis and connection to previous results from that study area. Thank you for your work! Abstract Line 16: Could you add some information about which area/time/well number you averaged the mean? And you did not mention the name or location of the study area in the abstract to which all numbers correspond to. Try to add this to make it more precise and enable the reader to set the study in space. Line 27: Mention that denitrification plays no major role in the study area. Otherwise diffuse recharge could be affected by this process. Introduction Line 37: Please add a few sentences why research for nitrate contamination is important. Line 63/64: The explanations "vadose (unsaturated)" and "groundwater (saturated zone)" could be earlier in the paragraph e.g. Line 38. Methods Line 107: In which depths are shallow, intermediate and deep groundwaters? Even more important than the screen length. Line 123: I did not check the paper, but how can the mean recharge stay the same, if 88% of the rates decrease? Because of highly positive outliers? Line 133-174: Really nice explanation of the method and its principles! Line 203: How strong was the relation between "Area of planted corn" and "fertilizer aplication rates"? R2? Should be really high as you substitute the Ninput mass by an area. Line 204: More information on the reduction- perhaps in bracktes "from... to..." or "by ….%" to estimate the effect (or its potential as marker in case of drastic drop). Line 230: I am not sure, how to imagine the "apparent" travel time as I only know about distributions (gamma or log-normal) of TTs. Your TT is the peak TT without any parts of it travelling faster or slower? So, you don't assume a mixed signal stemming from TTs from different ages (e.g. in 2010 10% signal/NO3 load from 1990, 40% signal from 1991, 50%...)? Line 234: Please, define shallow! Line 252-255: And the fertilizer input (Nsurplus) of 1990? Isn't this the most important input variable? Perhaps already cleared by Line 203, when adding R2. Line 263: "historical nitrate groundwater concentrations" or do you mean historical Ninput data? Results Line 292: I struggle to understand you differentiation between TTs and evolution of NO3. You don't use

NO3 as tracer to derive TTs and therefore you can correlate both? Or don't you use NO3 to derive transport rates? If you calculate one variable based on the other, isn't the correlation useless? Sorry for my confusion. You concept of TTs is quite different from ours. Line 332: Doesn't your canal leakage has also high NO3 from time to time, based on surface runoff from fertilized fields directly (pipes and drainages)? And can you add some information on the canal system previously? Is it also to drain the fields? Line 332: Why does influence of canals extends further from the canal? Isn't its influence decreasing with distance? Line 337: "nitrate reduction" add (also known as denitrification)? Line 338: "The partial dependence plot" add (Fig. 7) Line 342: I am surprised about your conclusion regarding the rapid aquifer response. You mention stratification and a groundwater age of 7years. Doesn't this account for a dampening of changing signals? Or what time do you assume with "rapid"? Or does this only correspond to the shallow, unstratified groundwater? Line 355: Do you have a recommendation how many data (stations) we need or how long time series should be to use your ML approach? Line 361: Isn't your "may be biased" a bit to optimistic? How can you distinguish a vanished NO3 imprint after denitrification from "stored somewhere in the upper soil"?

Figures Line 584: Is this pattern clockwise? Don't you need to switch the lower plots then? Line 594-595: Thanks for the explanation again! Line 597-600: Is there a difference between %inc and %Inc? It is not consistent in all figures. Line 622: Is there a space missing at "bData required further analyses"? Line 625: Why only "some models were ultimately based on <1049 obs"? According to your table all models fit the condition "<= 1049" and some "= 1049 observations".

---

## Short Comment (SC1) · 16 Jul 2020

One co-author on this manuscript is a USGS employee, and recent changes to their policy require that this discussion paper have the following information included: "This draft manuscript is distributed solely for purposes of scientific peer review. Its content is deliberative and pre-decisional. Because the manuscript has not yet been approved for publication by the U.S. Geological Survey (USGS), it does not represent any official USGS finding or policy."

Thank you

---

## Short Comment (SC2) · 16 Jul 2020

Initial response to Reviewers 1 and 2 regarding the manuscript titled: "Determination of vadose and saturated-zone nitrate lag times using long-term groundwater monitoring data and statistical machine learning".

In order to maintain clarity and organization, our responses have been uploaded to the supplement document titled "hess-2020-169-supplement.pdf", and should be available via the link below.

Thank you

Please also note the supplement to this comment:
https://www.hydrol-earth-syst-sci-discuss.net/hess-2020-169/hess-2020-169-SC2-supplement.pdf

**Supplement:**

Initial Response to Reviewers 1 and 2

**Determination of vadose and saturated-zone nitrate lag times using long-term groundwater monitoring data and statistical machine learning**

We are grateful to the reviewers for their thoughtful comments, which will improve the paper. Our initial responses to the first two reviews are indented below and shown in blue text.

We also note that the following disclaimer should be applied to the discussion paper:

**This draft manuscript is distributed solely for purposes of scientific peer review. Its content is deliberative and pre-decisional. Because the manuscript has not yet been approved for publication by the U.S. Geological Survey (USGS), it does not represent any official USGS finding or policy.**

**Reviewer 1**

Geology/Hydrogeology is missing. Provide a hydrogeological map, cross section, hydraulic characteristics of the aquifer etc.

> In the revised manuscript we will provide a cross section similar to those available in other publications focused on the Dutch Flats area. We will also add additional hydrogeological descriptions in the text.

Add more information about nitrate and its processes.

> In the revised Section 2.1 (Site Description) we will include more denitrification information, including more detail on findings from prior research in the area. Previous work suggests that denitrification is not extensive in the groundwater in this area.

Fig1.Change the maps. The figure needs to be more attractive. Add coordinate system.

> We will update the figure to include graticules. The figure includes a colored topographic map with appropriate symbology and detail necessary for the paper. We are uncertain what is meant by the suggestion to make the figure more attractive (e.g., overall figure should be changed?, improve resolution?, other?). We will also add a north-south vertical section showing the extent of the aquifer and schematic of groundwater flow directions.

The literature is out of date.

> We agree, as publication of machine learning models has recently been very rapid. We will update the manuscript with literature that has been published while the manuscript was in review.

Discuss the role of Nitrate isotopes for future contribution in this concept. Recent article provide the interaction between surface and groundwater bodies using nitrate isotopes which might be helpful in future works.

We are aware of some studies involving statistical approaches and N and O isotopes (e.g., https://doi.org/10.1002/2015WR018523; https://doi.org/10.1016/j.jconhyd.2015.07.003) but are unsure if these are the articles referred to by the reviewer.

In general, nitrate isotope ratios in the aquifer are fairly uniform (e.g., d15N = +4 ± 2 per mil) and consistent with recharge beneath fertilized agricultural land elsewhere. Previous work indicated a possible minor downward increase in d15N, which could be related to different recharge sources or historical changes in fertilizer/manure ratios. Evidence of denitrification (from dissolved gases and isotopes) was mostly limited to some of the deepest wells near the bottom of the aquifer. The effect of major canal leakage is considered largely to be nitrate dilution (i.e., relatively little nitrate addition, at least from the upgradient canals). Additional isotope data might be useful for documenting temporal shifts in recharge sources, or irrigation return flows to the river; however, it is difficult to know exactly the location or size of the contributing area for each well, especially the deeper ones. We will clarify some of these points, though a detailed discussion likely is beyond the scope of this paper.

**Reviewer 2 (Scott Gardner)**

The study presents the environmental setting well in terms of soil, climate, and land use, how-ever, more specific information (cross-sections or maps) on the geologic setting would be useful in evaluating spatial variability in lag times.

In the revised manuscript we will provide a cross section similar to those available in other publications focused on the Dutch Flats area. We will also add additional hydrogeological descriptions in the text.

The distance between the monitoring wells evaluated and the screens that are sampled to the sources of nitrate (probably fields) are not touched on in the manuscript and might be useful in explaining variance in lag times. Perhaps land use might also be important to consider nearby the wells, as interception, evapotranspiration, and other land use specific processes could be relevant to nitrate lag times.

Thank you for pointing this out. We do note some general trends over larger spatial areas, where wells north (upgradient) of the canals are lower in nitrate due to the absence of row crop production. The vast majority of wells are surrounded by agricultural fields, and we are lacking detailed year-to-year records of fertilizer application or crop production. We do focus in the paper on the proximity of wells to irrigation canals, which have been shown in past work to substantially impact groundwater nitrate concentrations due to focused recharge of lower-nitrate groundwater. We will add a couple additional sentences to the manuscript to expound on this information.

line 17 - I am not sure you need to include the part about it not being common to have unsaturated velocities slower than saturated, this has been the case in other studies and is not out of the ordinary (fractured bedrock aquifers, karst, etc.)

We agree that there are environments where this might be expected. We will clarify that this statement is a generalization for unconsolidated surficial aquifers receiving distributed recharge.

line 79 - perhaps provide a reference explaining the importance of canals in the region for readers that are not familiar with the study area.

> Although documented extensively elsewhere, we will insert a brief comment to emphasize the importance of the canals. The impact of canals will also be illustrated in a new figure summarizing the hydrologic setting. Thank you for pointing this out.

line 107 - here and everywhere after it is not clear what is meant by screen length, is this the depth bgs that the screen begins, or the size of the screen?

> In the revised manuscript we will define this as "length of screened interval."

please clarify line 157 - what is meant by 'bootstrapped' readers which are unfamiliar with computer science jargon may have trouble with this please clarify.

> In the revised manuscript we will define this term.

line 234 - what was the reasoning behind selecting 1 standard deviation for an acceptable range of results? If this selection was arbitrary then it should be made clear.

> In the revised manuscript we will note that the range based on 1 standard deviation was considered a reasonable range of recharge rates that might be considered based on prior research in the area.

figure s1 please change the colours on the nitrate concentrations to better contrast the results

> Figure S1 will be updated to provide more distinction between the different results.

---

## Short Comment (SC3) · 24 Jul 2020

Dear Reviewers,

Once again, in order to maintain clarity and organization, our responses have been uploaded to the supplement document, linked below.

The document contains our response to Reviewers 1, 2, and 3. Please note, our response to Reviewers 1 and 2 have remained unchanged from the previous discussion post. However, this document serves to incorporate Reviewer 3's comments, and com-

[Figure]

pile all discussion points into one document.

Thank you.

Please also note the supplement to this comment:
https://hess.copernicus.org/preprints/hess-2020-169/hess-2020-169-SC3-supplement.pdf

---

## Author Response (AR1)

**Response to Reviewers**

**Determination of vadose and saturated-zone nitrate lag times using long-term groundwater monitoring data and statistical machine learning**

We are grateful to the reviewers for their thoughtful comments, which have significantly improved the manuscript. Our initial responses to the reviewers are indented below and shown in blue text. These are followed by red text, which described the actual changes made to the manuscript and the locations of those changes. Line numbers refer to locations in the clean (unmarked) revised version of the manuscript.

We note that an additional review was provided by Christopher Green (USGS – Menlo Park, CA) for the purpose of internal USGS review. Some minor changes in the revised text reflect responses to this review.

We also note that, per USGS requirements, the following disclaimer should be applied to the **discussion paper** that is permanently posted on the HESS website, if at all possible.

**This draft manuscript is distributed solely for purposes of scientific peer review. Its content is deliberative and pre-decisional. Because the manuscript has not yet been approved for publication by the U.S. Geological Survey (USGS), it does not represent any official USGS finding or policy.**

**Reviewer 1**

Geology/Hydrogeology is missing. Provide a hydrogeological map, cross section, hydraulic characteristics of the aquifer etc.

> In the revised manuscript we will provide a cross section similar to those available in other publications focused on the Dutch Flats area. We will also add additional hydrogeological descriptions in the text.

> Figure 2 was added to show a cross section and text was added in Line 81. The cross section includes water table elevation and major geologic features. One transect of nested wells is also depicted to illustrate the distribution of wells and screen locations that are typical for the study area. The location of the cross section is also shown in map view in Figure 1.

Add more information about nitrate and its processes.

> In the revised Section 2.1 (Site Description) we will include more denitrification information, including more detail on findings from prior research in the area. Previous work suggests that denitrification is not extensive in the groundwater in this area.

> Lines 137-149 now address some factors affecting nitrate concentrations including the potential for denitrification to removed nitrate from groundwater.

Fig1.Change the maps. The figure needs to be more attractive. Add coordinate system.

> We will update the figure to include graticules. The figure includes a colored topographic map with appropriate symbology and detail necessary for the paper. We are uncertain what is meant by the suggestion to make the figure more attractive (e.g., overall figure should be changed?, improve resolution?, other?). We will also add a north-south vertical section showing the extent of the aquifer and schematic of groundwater flow directions.

> We created a new site map (Figure 1) to improve resolution and clarity, including features described above. Figure 1 also shows the location of the cross section (Figure 2).

The literature is out of date.

> We agree, as publication of machine learning models has recently been very rapid. We will update the manuscript with literature that has been published while the manuscript was in review. Recent publications have been added, especially in the Introduction and in our discussion of potential future work.

> In the revised manuscript more recent (2019-2020) references have been added in Lines 58-59 and 392.

Discuss the role of Nitrate isotopes for future contribution in this concept. Recent article provide the interaction between surface and groundwater bodies using nitrate isotopes which might be helpful in future works.

> We are aware of some studies involving statistical approaches and N and O isotopes (e.g., https://doi.org/10.1002/2015WR018523; https://doi.org/10.1016/j.jconhyd.2015.07.003) but are unsure if these are the articles referred to by the reviewer.

> In general, nitrate isotope ratios in the aquifer are fairly uniform (e.g., d15N = +4 ± 2 per mil) and consistent with recharge beneath fertilized agricultural land elsewhere. Previous work indicated a possible minor downward increase in d15N, which could be related to different recharge sources or historical changes in fertilizer/manure ratios. Evidence of denitrification (from dissolved gases and isotopes) was mostly limited to some of the deepest wells near the bottom of the aquifer. The effect of major canal leakage is considered largely to be nitrate dilution (i.e., relatively little nitrate addition, at least from the upgradient canals). Additional isotope data might be useful for documenting temporal shifts in recharge sources, or irrigation return flows to the river; however, it is difficult to know exactly the location or size of the contributing area for each well, especially the deeper ones. We will clarify some of these points, though a detailed discussion likely is beyond the scope of this paper.

> Lines 137-149 provides a little more background on isotopes as one line of evidence for investigating denitrification. However, available isotope data are much fewer than nitrate concentration data in this study (likely similar to many other monitoring networks) so it is difficult to say how nitrate isotope data would contribute to this particular modeling effort.

**Reviewer 2 (Scott Gardner)**

The study presents the environmental setting well in terms of soil, climate, and land use, how-ever, more specific information (cross-sections or maps) on the geologic setting would be useful in evaluating spatial variability in lag times.

> In the revised manuscript we will provide a cross section similar to those available in other publications focused on the Dutch Flats area. We will also add additional hydrogeological descriptions in the text.

> Figure 2 was added to show a cross section and text was added in Line 81. The cross section includes water table elevation and major geologic features. One transect of nested wells is also depicted to illustrate the distribution of wells and screen locations that are typical for the study area. The location of the cross section is also shown in map view in Figure 1.

The distance between the monitoring wells evaluated and the screens that are sampled to the sources of nitrate (probably fields) are not touched on in the manuscript and might be useful in explaining variance in lag times. Perhaps land use might also be important to consider nearby the wells, as interception, evapotranspiration, and other land use specific processes could be relevant to nitrate lag times.

> Thank you for pointing this out. We do note some general trends over larger spatial areas, where wells north (upgradient) of the canals are lower in nitrate due to the absence of row crop production. The vast majority of wells are surrounded by agricultural fields, and we are lacking detailed year-to-year records of fertilizer application or crop production. We do focus in the paper on the proximity of wells to irrigation canals, which have been shown in past work to substantially impact groundwater nitrate concentrations due to focused recharge of lower-nitrate groundwater. We will add a couple additional sentences to the manuscript to expound on this information.

> In the revised manuscript, text was added in Lines 76-77, 91-93, 108-111 to emphasize the potential influence of canals and surface irrigation on groundwater in the study area. We also emphasize that crop fields are present across most of the study area and cite Figure S2 as an illustration (Lines 353-354).

line 17 - I am not sure you need to include the part about it not being common to have unsaturated velocities slower than saturated, this has been the case in other studies and is not out of the ordinary (fractured bedrock aquifers, karst, etc.)

> We agree that there are environments where this might be expected. We will clarify that this statement is a generalization for unconsolidated surficial aquifers receiving distributed recharge.

> The sentence was removed from the abstract, clarified, and reinserted on Lines 341-342.

line 79 - perhaps provide a reference explaining the importance of canals in the region for readers that are not familiar with the study area.

Although documented extensively elsewhere, we will insert a brief comment to emphasize the importance of the canals. The impact of canals will also be illustrated in a new figure summarizing the hydrologic setting. Thank you for pointing this out.

An existing statement on Lines 84-86 discussed the importance of canals to the region. Additional text was added to Lines 108-111.

line 107 - here and everywhere after it is not clear what is meant by screen length, is this the depth bgs that the screen begins, or the size of the screen?

In the revised manuscript we will define this as "length of screened interval."

This change was made in Lines 114-119.

please clarify line 157 - what is meant by 'bootstrapped' readers which are unfamiliar with computer science jargon may have trouble with this please clarify.

In the revised manuscript we will define this term.

The definition was added in Line 183.

line 234 - what was the reasoning behind selecting 1 standard deviation for an acceptable range of results? If this selection was arbitrary then it should be made clear.

In the revised manuscript we will note that the range based on 1 standard deviation was considered a reasonable range of recharge rates that might be considered based on prior research in the area.

This clarification was added in Lines 267-269.

figure s1 please change the colours on the nitrate concentrations to better contrast the results

Figure S1 will be updated to provide more distinction between the different results.

Figure S1 was completely re-worked to show the original data (Figure S1A) and the nitrate data adjusted for total travel time (Figure S1B). These figures are both cited in the main text.

**Response to Reviewer 3 (Sophie Ehrhardt)**

Abstract:

Line 16: Could you add some information about which area/time/well number you averaged the mean? And you did not mention the name or location of the study area in the abstract to which all numbers correspond to. Try to add this to make it more precise and enable the reader to set the study in space.

> We agree, this is good information to add. We will indicate that the mean was with respect to an area (i.e., the Dutch Flats area).
>
> This change was made in Lines 15-16.

Line 27: Mention that denitrification plays no major role in the study area. Otherwise diffuse recharge could be affected by this process.

> In the revised manuscript we will mention in the abstract the lack of suggested denitrification.
>
> Text added to Line 24 addresses this comment.

Introduction:

Line 37: Please add a few sentences why research for nitrate contamination is important.

> We feel this material has been heavily documented in nitrate-related research already published – many of which referenced in this paper – and well known by the readers. However, some general introductory sentences were added.
>
> Text was added in Lines 37-39.

Line 63/64: The explanations "vadose (unsaturated)" and "groundwater (saturated zone)" could be earlier in the paragraph e.g. Line 38.

> In the revised manuscript we will provide these synonyms in the first paragraph of the introduction.
>
> The synonyms were added in Line 41 (4th sentence of the introduction).

Methods:

Line 107: In which depths are shallow, intermediate and deep groundwaters? Even more important than the screen length.

> We agree the actual depths are important information. We will add an additional sentence with the range of vadose zone and saturated zone (depth below water table) thicknesses in this study. This will complement the hydrogeologic cross-section, which we will add in response to Reviewers 1 and 2.

Example well depths are now depicted in Figure 2 and additional details are included in Lines 114-119.

Line 123: I did not check the paper, but how can the mean recharge stay the same, if 88% of the rates decrease? Because of highly positive outliers?

In previous work, the recharge rates were slightly lower in the majority of wells, but the overall mean recharge rate was not statistically different.

No changes were made in the revised document.

Line 203: How strong was the relation between "Area of planted corn" and "fertilizer application rates"? R2? Should be really high as you substitute the Ninput mass by an area.

This is a good point. As discussed later in this paragraph, we were not simply substituting a proxy (area of planted corn) for actual fertilizer data. The choice we had to make was between a proxy and "no data" for years prior to 1987. Although the correlation was low for more recent years ($R^2 = 0.26$), groundwater nitrate concentrations have been closely linked to the area of row crops, including corn, in numerous water quality studies. The low correlation may be due to better fertilizer management with agricultural producers applying less fertilizer per hectare than in the past. As a result, we felt this was our best choice for incorporating an important dynamic variable into the study.

Discussion of this issue was expanded in the text between Lines 229-236.

Line 204: More information on the reduction- perhaps in brackets "from... to..." or "by . . ..%" to estimate the effect (or its potential as marker in case of drastic drop).

Thank you, this is a good location to give a sense of the magnitude of observed change. In the revised manuscript we will add quantitative information for fertilizer and planted corn, respectively.

The magnitude of changes is noted in more detail in Lines 229-233.

Line 230: I am not sure, how to imagine the "apparent" travel time as I only know about distributions (gamma or log-normal) of TTs. Your TT is the peak TT without any parts of it travelling faster or slower? So, you don't assume a mixed signal stemming from TTs from different ages (e.g. in 2010 10% signal/NO3 load from 1990, 40% signal from 1991, 50%...)?

We use the term "apparent" and mentioned imperfect age-dating tracers to address this exact question, which is that a single groundwater age typically represents a mean age reflecting the different recharge year for each water molecule in sample. The equations we present are simplified representations (as are tracers) comparable to piston-flow assumptions (a common simplification when interpreting groundwater age-dating tracer data).

In Lines 256-261 we clarified that for short-screened wells such as the ones used in this study, the uncertainty (variability) in groundwater age is generally smaller than it might be in long-screen wells. Furthermore, it may be expected that regional changes in nitrate recharge fluxes will be smoothed over a period of years. Thus, our model assumption that individual samples represent approximately discrete travel times.

Line 234: Please, define shallow!

We can understand your frustration here. We will refine our descriptions as stated in the response to the Line 107 comment above. When the cross section is provided, it will show how the terms "shallow", etc, are tied more to depth below the water table than to total well depth.

Example well depths in Figure 2 and additional text in Lines 114 – 119 were added for clarification.

Line 252-255: And the fertilizer input (Nsurplus) of 1990? Isn't this the most important input variable? Perhaps already cleared by Line 203, when adding R2.

We agree, the fertilizer input certainly would have been a very beneficial variable to include; though, we unfortunately did not have enough data to include this variable in the analysis. Line 275 – 284 discusses dynamic variables and acknowledges stronger dynamic predictors could provide for an interesting follow up study. We will add to this section (i.e., Lines 275 – 284), specifically calling out N loading as a factor to consider in future studies, although these data are very difficult to reconstruct for long-term studies.

Additional discussion of the potential benefit of a well-defined nitrate input function is included in Lines 409-411. We also note (just for discussion here, not in the manuscript) that in an ideal case, we would have a very well-defined input function *and* long-term time series of groundwater nitrate concentrations. In this case, perhaps recurrent neural networks or other models suitable for time series data could be used for similar study. Unfortunately, we have neither of these data sets. The intermittent sampling of wells is part of the reason we chose the Random Forest model, as traditional time series analyses (or machine learning approaches that leverage time series data) were not suitable for these data.

Line 263: "historical nitrate groundwater concentrations" or do you mean historical Ninput data?

Historical groundwater nitrate concentrations are correct here. We unfortunately did not have long-term Ninput data to use for this study.

No change was made to the text in response to this comment. However, we do think the Figures S1A and S1B are interesting and relevant when considering connections between historical nitrate groundwater concentrations and historical N inputs.

Results:

Line 292: I struggle to understand your differentiation between TTs and evolution of NO3. You don't use NO3 as tracer to derive TTs and therefore you can correlate both? Or don't you use NO3 to derive transport rates? If you calculate one variable based on the other, isn't the correlation useless? Sorry for my confusion. You concept of TTs is quite different from ours.

The TT was not calculated based on nitrate, but rather the vertical vadose and saturated zone distance at each well. The rationale was that there is a known relationship between long travel times and low nitrate, and short travel times and high nitrate. Then, we used the Random Forest model to determine which TT had the largest influence on the model's overall ability to predict nitrate concentrations. Put another way:

- Total travel time was estimated for each well as a function of site characteristics (e.g. vadose zone depth) and saturated/unsaturated transport rates.

- Transport rates were varied across specified ranges such that alternative total travel times were constructed for each well

- The model was re-trained and tested for each set of alternative total travel times

- Permutation importance (measured as % increase in MSE or %IncMSE) was calculated for each re-trained model. When calculating permutation importance for total travel time, we are randomly shuffling the total travel time observations across all of the wells to essentially ruin the structure of the dataset. The model is run with this shuffled version of the dataset, and we document that change in error that occurs for the model run with the shuffled data vs. the original correctly-structured data.

- When %IncMSE was high, this indicated the model was sensitive to changes in total travel time.

- The permutation importance of total travel time (%IncMSE) varied depending on the transport rate values used to calculate the total travel time.

- The greatest %IncMSE occurred when the vadose zone transport rate was 3.5 m/yr, and saturated zone transport rate was ~3.7 m/yr. Therefore, we concluded that these were the optimal transport rates for the RF model.

Figure S1B, was added to the Supplemental Information to help illustrate the model outputs, which are consistent with other "reconstructed" input histories for groundwater nitrate (e.g., Puckett et al. 2011). More broadly, we used a machine learning model and appropriate diagnostic tools (%$MSE_{inc}$, partial dependence plots, etc.) to determine whether the models were reasonable, and we demonstrate that the model captured processes that are consistent with conceptual understanding of the hydrologic system. We also used an independent metric (i.e., independent of our field data; %$MSE_{inc}$) to select the "optimal" model (and therefore optimal transport rates) and those results were consistent with previous field data. Additional text was also added to Lines 161-164

Line 332: Doesn't your canal leakage has also high NO3 from time to time, based on surface runoff from fertilized fields directly (pipes and drainages)? And can you add some information on the canal system previously? Is it also to drain the fields?

Previous studies found that when water was flowing through the Interstate Canal (largest canal in this region), nitrate concentrations were less than 0.06 mg N L-1, and did not exhibit large spikes, during their collection period, in nitrate concentrations. Below is an excerpt from Böhlke et al. (2007) showing the nitrate concentrations in the Interstate Canal to be very low (data collected over a 4-year period with seasonal irrigation flow peaks).

[Figure]

While some of the smaller ditches could indeed carry tailwater, the major canals in this region serve as the primary delivery (only) canals in the region. We plan to add additional information regarding the dependence this region has on canals.

The nitrate concentrations for canal water from previous work were already included in the original manuscript (Line 104 in this revised version).

Line 332: Why does influence of canals extends further from the canal? Isn't its influence decreasing with distance?

Thank you for pointing this out, as the wording is not completely clear. The text was intending to state that the influence from canal leakage is exhibited further from major canals than minor canals. We will adjust the text to state: "The effects of minor and major canals, where groundwater [NO$_3$-] in the vicinity of canals is diluted by canal leakage, and the influence of major canals extends further from the canal when compared to minor canal results."

Line 372-373 was changed to state: "the effects of minor and major canals, where groundwater [NO3-] in the vicinity of canals is diluted by canal leakage, and the influence of major canals extends a longer distance when compared to that of minor canals"

Line 337: "nitrate reduction" add (also known as denitrification)?

Correct, and per comments from Reviewer 1 and 2, we will be incorporating additional discussion and information into the manuscript related to denitrification.

The wording was changed to "denitrification" in Line 377.

Line 338: "The partial dependence plot" add (Fig. 7)

In the revised manuscript will add "(Fig. 7)" to the text currently on Line 338.

Figure 7 is now Figure 8, and this comment is addressed on Line 378 of the revised manuscript.

Line 342: I am surprised about your conclusion regarding the rapid aquifer response. You mention stratification and a groundwater age of 7years. Doesn't this account for a dampening of changing signals? Or what time do you assume with "rapid"? Or does this only correspond to the shallow, unstratified groundwater?

Our reference point for the term "rapid" is the many previous age-dating studies in shallow unconfined aquifers in agricultural areas where the mean transit time, and therefore the groundwater quality response time, in the aquifer is "decades". As noted earlier in the paper, the Random Forest model may be strongly influenced by younger groundwater with more pronounced nitrate signals.

Reworded to clarify on Lines 381-383

Line 355: Do you have a recommendation how many data (stations) we need or how long time series should be to use your ML approach?

Hard to make a recommendation here, but certainly the larger the dataset (and number of stations), the better. Larger datasets provide more data used to train each tree, ultimately giving each tree more data to "learn" from, making the overall forest more robust. Additional data would also help to ensure that the full range of observations are captured in the dataset. Future research could compare results by taking various subsets of the complete data set to provide insight on data requirements.

Lines 444-445 discusses this as a need for future studies to research.

Line 361: Isn't your "may be biased" a bit to optimistic? How can you distinguish a vanished NO3 imprint after denitrification from "stored somewhere in the upper soil"?

This is a good point. We will add that vertical sampling of the vadose zone for nitrate would provide ideal data to address whether this approach "misses" nitrate stored in the unsaturated zone.

Line 401-404 addresses this issue. We also note that denitrification could also occur in the vadose zone.

Figures

Line 584: Is this pattern clockwise? Don't you need to switch the lower plots then?

In the revised manuscript the text will reflect the correct order of the plots.

The caption for Figure 3 has been re-written. Letters were also added to help distinguish each plot.

Line 597-600: Is there a difference between %inc and %Inc? It is not consistent in all figures.

There is no difference intended, but the revised manuscript will be updated to maintain a consistent nomenclature for this between the text and figures. Thank you for pointing this out.

This has been done throughout the paper.

Line 622: Is there a space missing at "bData required further analyses"?

Thank you for your attention to detail – the table will be updated to maintain a consistent format.

Foot note to Table 1 has been corrected.

Line 625: Why only "some models were ultimately based on <1049 obs"? According to your table all models fit the condition "<= 1049" and some "= 1049 observations".

Table 2 reflects the further analyses that were performed on the model when the dynamic predictors were included in the analysis. In the revised manuscript we will add a comment to ensure that readers are aware this table is for the analysis that included dynamic variables. The reason some of the models included <1049 observations is due to the limitation in historical dynamic variable data available, where some data were not present prior to 1946. Therefore, the number of observations were decreased for some of the slower transport rates that result in a total travel time prior to 1946.

For example, if a sample was collected in 2000, and it had a 60-year total travel time, the dynamic variable would be assigned a value from 1940. However, the dataset was limited to 1946, so any observation assigned a dynamic variable year prior to 1946 had to be excluded.

This is now clarified in Line 712-713 (footnotes of Table 2).

[revised manuscript text omitted]
 = 1,049), differentiated by well depth labels. Note that well depth is determined within individual well nests, and therefore not an absolute indicator of well depth from the land surface (see discussion in main text). (B) Plot of nitrate data adjusted for total travel time when calculated using optimal transport rates identified in the study. The adjusted nitrate data are the modelled input of nitrate to the system, starting at the land surface (infiltration). The overall pattern of input over time is similar to other studies where nitrate data were adjusted for groundwater age (e.g., Puckett et al. 2011) although inputs are more typically plotted based on recharge year rather than infiltration year.

**Commented [TG1]:** Figure S1 and caption were reworked.

[Figure]

**Figure S2:** Visual comparison of 1999 Dutch Flats center pivot irrigated fields to 2017 center pivot irrigated fields using NAIP, NAPP, (USDA 2017) and LANDSAT imagery (USGS 2017) (Table S1). Sample sites shown in the figure are from a representative subset of wells selected for comparison in the Wells et al., 2018 study.

[Figure]

**Figure S3:** Center Pivot Irrigated Area based on observations from aerial imagery. The years 1999 and 2003 were used as breaks for determining linear regression equation.

**Table S1:** Years when aerial imagery was utilized to digitize center pivot irrigated fields.

| Year | Estimated Center Pivot Irrigated Area (hectare) |
|---|---|
| 1975 | 429 |
| 1990 | 2507 |
| 1995 | 3015 |
| 1999* | 3830 |
| 2001 | 5685 |
| 2003* | 7361 |
| 2004* | 7804 |
| 2005* | 8341 |
| 2006* | 8822 |
| 2010* | 10577 |
| 2014* | 13591 |
| 2017 | 14253 |

*Years analysed with NAIP or NAPP imagery

---

## Author Response (AR2)

**Response to Reviewers**

**Determination of vadose and saturated-zone nitrate lag times using long-term groundwater monitoring data and statistical machine learning**

We are very grateful to Pia Ebeling for a detailed and helpful review of the paper. We are also grateful to the second reviewer, who reviewed the manuscript for the second time and had no further revisions to suggest.

The marked, revised manuscript is appended below.

In our responses, we tried to balance the new reviewer suggestions with those of previous reviewers as well as the overall genesis of the manuscript. We made one significant change (rearranging Sections 2.3-2.5 of the Methods), and several minor changes in line with reviewer comments. We were less responsive to the overriding theme of shortening the Methods section. We agree that it is possible to cut out some of the text in this paper and replace with references. At the same time, although machine learning (and RF in particular) is becoming much more common, there are still many hydrologists and practitioners who are unfamiliar with the methods (Saia et al., 2020). At least one reviewer specifically noted their appreciation for the additional detail. Finally, as an author group, we believe it is common for those who are intimately familiar with the study site (unlikely, for this particular site) and/or the methods used, to skim or skip the Methods section, read the Results and Discussion, and return to the methodological details to resolve specific questions that arise. We did reduce redundancy as suggested and feel the Methods section is of reasonable length in its revised form.

In the Results and Discussion, the points raised were consistent with other reviewers and involved critical limitations that must be emphasized. Pia's review helped us identify where and how we should strengthen our discussions around the exploratory/limiting aspects of the study. We believe the discussion of limitations is even more robust after revisions.

Saia, SM, NG Nelson, AS Huseth, K Grieger, and BJ Reich. 2020. Transitioning Machine Learning from Theory to Practice in Natural Resources Management. *Ecological Modelling* 435: 109257. https://doi.org/10.1016/j.ecolmodel.2020.109257.

**Reviewer 2, Pia Ebeling**

This study uses an interesting innovative approach to estimate transport rates and lag time of NO3 in the unsaturated and saturated zone from groundwater NO3 data using random forests. However, the manuscript could be better streamlined according to the messages; especially the method chapter is quite lengthy, repetitive and hard to understand (it is longer than results and discussion). This unfortunately also hampers a clear understanding of the results and discussion chapter in some parts. Some concrete suggestions are given in the detailed comments, though they are not complete and the method sections needs to be revised. The framework is promising and could replace expensive isotope sampling, therefore, I suggest publication after including the requested changes.

Major concerns

Why do you use only vertical velocities? For the unsaturated zone, this concept is common, but I am uncertain why this is used for the saturated zone as well. I did not understand the assumption of shallow groundwater in this context (L.253). Please, clarify this more. Why would it be important to know travel times in vertical dimension without horizontal component? How can the flow be dominantly vertical if the aquifer bottom is considered a no-flow boundary? I think this limitation should also be discussed in the discussion.

AR: This is an important concept, for which we added some text in the Methods (Section 2.4, Lines 214-224). As in many other surficial (water-table) aquifers, we assume groundwater flow is largely horizontal, but with a consistently downward component, such that ground water ages (travel times since recharge) increase with depth. Acknowledging the study area does not represent an idealized surficial aquifer, we listed some general and local observations leading to the linear approximation of vertical groundwater age gradients in this setting. Nonetheless, we certainly agree there is substantial variability in the vertical velocities, which may have limited the performance of the simplified models.

Why do you use the importance of the total travel time to infer transport rates? This argument is not clear for me.

AR: The most direct answer is in the text (lines 303-305): "Total travel time also had the greatest variability in importance among the fifteen variables, with a range of 18.4% between the upper and lower values, suggesting some model sensitivity to lag times."

Abstract:

L 17: I do not understand this sentence, though I think I know what you mean. Consider reformulation.

AR: We deleted the word "contrasting" to improve the flow of the sentence.

Introduction:

L. 38-39 I do not understand this sentence. What do you mean by "responses … can be complicated by uncertainties"? Do you mean predictions of responses?

AR: We added the word "Predicting…" to the beginning of the sentence. Thank you for pointing this out.

L 48 Are lag times the same as groundwater ages for you?

AR: No. This sentence refers to both groundwater age-dating (i.e., saturated zone travel time) and vadose zone methods to quantify lag time. Edits for clarification were made in lines 47-48.

L 60-61. Why is the screen depth a proxy for lag times? Wouldn't this only be the case for homogeneous settings and only for vertical movement?

AR: We are referring in this specific sentence to previous studies that used well depth as a proxy for lag times. Observations of vertically stratified groundwater age (increasing age with depth) and nitrate concentrations in aquifers are very common in literature. Hydrologic theory suggests and age-dating studies have shown a strong vertical component to groundwater flow in the shallow parts of the groundwater system. In terms of groundwater age, that is manifested by a linear increase in groundwater age with depth below the water table. We acknowledge that deeper portions of the aquifer may have a greater horizontal component to flow (see additional discussion to comments about horizontal flow below).

Methods (74-290):

**AR overview on Methods section:** Our philosophy on Methods sections is that a reader who is already very familiar with the study site or well-versed in the methods can easily skip to the Results and Discussion section and return later to review methods if questions arise. As recently argued in Saia et al. (2020), machine learning papers warrant more descriptive methodology sections.

Section 2.1 reads very long-winded and not all detail is needed later in the discussion. Please, consider condensing this section.

AR: We appreciate the concern about brevity. However, this section has already been expanded with details in part based on previous reviewers' comments. Further, we feel the site description adds valuable context to the study.

L 114 What are nested wells? Wells with more than one screen?

AR: Yes, as described in Lines 112-119. There is one borehole with multiple tubes, with each tube having the screens at different depths.

Section 2.2 and 2.3: In my opinion, these two sections read quite unfocused and I think they could be condensed and merged into one section. E.g. L. 183-187 This seems unnecessary to explain as this is the principle of random forest which can be easily referenced from literature. Similarly, l 187-196 can be shortened into one sentence saying which approach was used to evaluate variable importance. Actually, 2.2. reads as a summary of what will be explained in 2.3-2.5, which is not necessary. I think you can easily remove it.

AR: Section 2.2 and 2.3 (Section 2.5 in the revised manuscript, after reorganization) were slightly modified to reduce redundancies.

L 169: This sentence seems redundant

AR: Section 2.3 (Section 2.5 in the revised manuscript, after re-organizing) was adjusted to remove redundancies created by this sentence.

L 170 Why did you use nested resampling? I do not see that you discuss those results later. If you used it for tuning, please specify this including the tuning method and parameters.

AR: We used five-repeated five-fold cross-validation to evaluate model performance sensitivity. The five repetitions were performed to account for variation in the data assigned to training/testing splits. To clarify this in the manuscript, the sentence was changed from "This process was repeated five times to create a total of 25 models, similar to the approach used by Nelson et al. (2018)." to "We repeated the five-fold cross validation process five times to create a total of 25 models, similar to the approach used by Nelson et al. (2018), in order to assess sensitivity of model performance to the data assigned to the training and testing folds."

L. 174 I would suggest the authors to be clearer as permutation importance and pdps are not used to evaluate the model performance but to interpret the model results.

AR: Revised to: "Permutation importance, partial dependence and Nash-Sutcliffe Efficiency (NSE) were quantified to evaluate model performance and to interpret results."

L 174-182 One sentence is enough to say that you use NSE to evaluate model performance.

AR: We deleted the sentence just above equation 1, which was redundant.

L 198 This sentence is vague and not informative. I would even say it is not correct actually. Within one pdp only one variable (x-axis) is varies while for the others mean values are used. Please reformulate or delete. In my opinion, this paragraph can also be shortened.

AR: Referenced sentence has been removed from the manuscript.

L 204-208 Can be merged into one sentence. It is not relevant to know that data were imported and clipped, just that you selected the "wells within the study area with a corresponding depth …" would be enough.

AR: We removed the comment about importing and clipping data.

L 233: I am sorry, I did not understand how you use the dynamic descriptors. "annual median [NO3] was assigned a lagged dynamic value", what exactly does that mean? And are the dynamic variables also variable in space?

AR: Dynamic variables were not variable in space as applied in this study. We now point this out as a potential future avenue of exploration (second paragraph of Section 3.1, line 311 in revised manuscript) The exact method for lagging dynamic predictors is given in an example (just below Equation 3, next to last paragraph in Section 2.4 in the revised manuscript).

L 267-269 this sentence in unnecessary like it is. You already state before what range of recharge rates you used. Or maybe the second part could be supported with references: what are those realistic mean values and why are they realistic?

AR: We revised to note in the first sentence that the recharge rates obtained varied by a factor of 4. Then deleted the second sentence as suggested.

L 264-280 Could the calculated parameter ranges of the two paragraphs not be easily presented in a table? It would increase the understanding and not read that lengthy. The text could be strongly shortened then.

AR: These paragraphs explain how we arrived at the range of transport rates. Most of the text would need to remain. In that case, the table would either repeat information in the sentences and/or would only contain two sets of values (the range of vadose-zone and the range of saturated-zone transport rates).

L 285-288 Why did you take this assumption? You could also have used various features derived from the dynamic predictors including mean values over several years. Considering the uncertainty in travel time estimates, I think this approach is not well supported. However, it is still slightly unclear to me how you used the dynamic values and linked them to the annual median [NO3]. Did you assign one past value to each well and total lag time (for one transport rate combination)? If so, why do think this is possible if we also have horizontal groundwater movement and the annual median [NO3] is not only dependent on vertical travel times at the well location but on vertical travel times somewhere else plus the horizontal travel times. Please, clarify. Maybe a conceptual figure would help to understand how you link dynamic descriptor values to certain wells and corresponding annual median [NO3].

AR: These lines are now located at approximately Lines 250-253 of the revised manuscript. We did assign one past value, as you describe (and as described in the text). It is possible that some other methods could yield results (other studies have used decadal nitrate to roughly estimate travel times, as cited in the introduction, Lines 57-66). Regarding horizontal groundwater flow, please see overview discussion in response to Major Concerns, above.

L 289-293 This part is jumping back to RF application. I think the method section should be restructured with the concept of travel times being presented first and then the RF application used to predict the travel times.

AR: We reorganized the methods sections (2.3 through 2.5) as suggested. The RF details are now in Section 2.5.

Results and Discussion (L. 297- 425)

L 299 Which is the "initial model"? I cannot follow. This should be stated more clearly in the methods.

AR: This was clarified using a parenthetical reference "(using both static and dynamic predictors)".

L 325-335 This paragraph belongs to the methods. Why do you use the variable importance of total travel times to determine the "optimal" transport rates? I struggle to understand why this approach is promising.

AR: We believe this paragraph belongs in the current location. The information presented here naturally follows text describing the initial round of modelling, which was discussed in the previous section to the paragraph in question. As noted in the second sentence of Section 3.2, we noted substantial variation in total travel time %$_{inc}$MSE. We also note in Section 3.3, bullet #3, the opportunities and limitations to this approach. We are aware of, and discuss, the fact that this approach may not be successful for other studies.

L. 338 Do you think these ratios are transferable to other locations? I would expect this to be site specific. This is not clearly stated here, as the statement sounds rather general.

AR: Thank you for pointing this out. We think the general concept of such ratios could be valuable elsewhere, but these values are specific to the study area. We revised the text to indicate this.

L 345 "second analysis"? What was the paragraph before then? This needs to be specified in the methods.

AR: The paragraph above (first paragraph of Section 3.2) describes the "second analysis". To clarify, we added a parenthetical reference ("(using only static variables") to this sentence.

L 346 How uncertain do you see those values considering the equifinality within the Vs/Vu ratios shown in Fig. 6? Is it meaningful to select one model in this case? Are there more ways to constrain the equifinality problem except data of recharge rates?

AR: While equifinality is an issue with many modelling approaches, we felt the quantitative analysis and subsequent results of the single rate and band of recharge rate ratios compared favorably enough with observed rates that it warranted discussion of both. However, we acknowledge there is considerable uncertainty, and thus our call for further testing and the use of other data to compare with ML model outcomes.

L 355. "mean recharge value derived from groundwater ages in intermediate wells (1.22 m/yr, n = 13)." Please provide the reference here.

AR: References have been added (approximately line 355 of the revised version)

3.3 Section: What data or prior knowledge would someone need to apply your new approach to estimate lag times? Would you trust to apply the method without having groundwater ages to validate?

AR: Starting with the abstract of the paper, we highlight limitations of the study and the need for corroboration of results "with a robust conceptual model and complementary information such as groundwater age." Thus, a robust conceptual model is an essential starting point. In Section 3.3, bullet #3, we state "Testing the approach of using %$_{inc}$MSE in other vadose and saturated zones, with substantial comparison to previous transport rate estimates, is warranted." Thus, we feel groundwater age or some other comparisons are still necessary when using the approach. This is reinforced by Section 3.3 Opportunities and Limitations…, where four out of five bullets include detailed discussion of model limitations, need for further testing at other sites with different recharge and redox patterns, and other data that could potential guide a similar modeling process.

Do you think this method also works in areas with higher denitrification impact where NO3 is less conservatively transported? Could the approach be hampered if denitrification potential or other subsurface conditions are heterogeneous (e.g. linked to hot spots such as pyrite lenses or hydraulic conductivity)?

AR: In the second bullet of Section 3.3, we broadened the discussion of nitrate extinction depths to include "heterogeneity in denitrification potential".

Conclusion:

L 440 What do you mean by "comparisons of data-driven analyses with complementary datasets"

AR: Added examples to the text to help clarify. Text now reads" "… comparisons of data-driven analyses with complementary datasets and/or modelling (e.g., field-based recharge rate estimates, finite-difference flow model)"

Figures:

L 686:"Vs/Vu"

AR: Thank you for catching this typo. It has been fixed.

[revised manuscript text omitted]